 **eLIFE**

# Disordered clusters of Bak dimers rupture mitochondria during apoptosis

Rachel T Uren[1,2], Martin O'Hely[1,2†], Sweta Iyer[1,2], Ray Bartolo[1‡], Melissa X Shi[1,2], Jason M Brouwer[1,2], Amber E Alsop[1,2], Grant Dewson[1,2], Ruth M Kluck[1,2*]

[1]The Walter and Eliza Hall Institute of Medical Research, Parkville, Victoria, Australia; [2]Department of Medical Biology, The University of Melbourne, Parkville, Victoria, Australia

**Abstract** During apoptosis, Bak and Bax undergo major conformational change and form symmetric dimers that coalesce to perforate the mitochondrial outer membrane via an unknown mechanism. We have employed cysteine labelling and linkage analysis to the full length of Bak in mitochondria. This comprehensive survey showed that in each Bak dimer the N-termini are fully solvent-exposed and mobile, the core is highly structured, and the C-termini are flexible but restrained by their contact with the membrane. Dimer-dimer interactions were more labile than the BH3:groove interaction within dimers, suggesting there is no extensive protein interface between dimers. In addition, linkage in the mobile Bak N-terminus (V61C) specifically quantified association between dimers, allowing mathematical simulations of dimer arrangement. Together, our data show that Bak dimers form disordered clusters to generate lipidic pores. These findings provide a molecular explanation for the observed structural heterogeneity of the apoptotic pore.

*For correspondence: kluck@ wehi.edu.au

**Present address:** [†]Barwon Infant Study, Deakin University, Geelong, Australia; [‡]Murdoch Children's Research Institute, The Royal Children's Hospital, Parkville, Australia

**Competing interests:** The authors declare that no competing interests exist.

## Introduction

The Bcl-2 family of proteins are the principal regulators of apoptotic cell death, with either Bak or Bax required to permeabilise the mitochondrial outer membrane (*Lindsten et al., 2000*; *Wei et al., 2001*). Bak and Bax are triggered to convert to their activated conformation by the binding of BH3-only relatives, such as Bid and Bim (*Gavathiotis et al., 2010*; *Dai et al., 2011*; *Du et al., 2011*; *Czabotar et al., 2013*; *Leshchiner et al., 2013*; *Brouwer et al., 2014*), and once activated can be sequestered by pro-survival relatives, such as Mcl-1 and Bcl-x$_L$ (*Llambi et al., 2011*).

The ability of Bak and Bax to change conformation and oligomerise in the mitochondrial outer membrane appears crucial for their capacity to perforate this membrane (*Westphal et al., 2014*). Both proteins comprise nine α-helices that adopt a globular fold in the non-activated state (*Suzuki et al., 2000*; *Moldoveanu et al., 2006*). Non-activated Bak is anchored in the mitochondrial outer membrane via the α9-helix forming a transmembrane domain (*Iyer et al., 2015*). In contrast, non-activated Bax is largely cytosolic due to binding of α9 to its own hydrophobic groove (*Wolter et al., 1997*; *Gahl et al., 2014*). Following binding of BH3-only proteins, both Bak and Bax undergo similar conformation changes including α1 dissociation (*Weber et al., 2013*; *Alsop et al., 2015*) and separation of the 'latch' domain (α6-α8) (*Czabotar et al., 2013*; *Brouwer et al., 2014*) from the 'core' domain (α2-α5). The core and α6 then collapse onto the membrane surface and become shallowly inserted into the membrane to lie in-plane (*Aluvila et al., 2014*; *Bleicken et al., 2014*; *Westphal et al., 2014*). Symmetric homodimers then form when the exposed BH3 domain of each molecule is re-buried in the hydrophobic groove of another activated molecule (*Dewson et al., 2008, 2012*; *Czabotar et al., 2013*; *Brouwer et al., 2014*). Bax oligomers may also form prior to membrane insertion (*Luo et al., 2014*; *Sung et al., 2015*).

**eLife digest** A healthy organism must carefully remove unwanted, diseased or damaged cells. These unwanted cells can bring about their own death in a controlled process known as apoptosis. Maintaining an appropriate level of apoptosis is crucial to good health: excessive cell death can contribute to neurodegenerative disorders, whereas too little can result in cancer.

All cells contain powerhouses called mitochondria, which produce energy. Mitochondria are the scene of a critical 'point of no return' in apoptosis. When a cell receives a death signal, a 'killer' protein known as Bak punches holes (or pores) in the membrane of the mitochondria. These pores allow toxic molecules to leak out from the mitochondria into the interior of the cell, where they trigger a series of events that dismantles the cell from the inside out.

To create the pores, Bak undergoes extensive shape changes that allow the proteins to form dimers that then cluster and perforate the membrane. To investigate how Bak clusters assemble on the mitochondrial membrane, Uren et al. used cultured cells and biochemical techniques to show where the Bak dimers contacted each other before and after the pore formed; these findings were complemented with mathematical modelling. The results show that during apoptosis, Bak dimers contact each other at several different places (rather than at one or two places) to assemble into disorderly, ever-changing clusters. Based on these observations, Uren et al. suggest that the enlarging clusters stress the membrane and cause pores to form.

The next step is to investigate whether physical forces that act within the mitochondrial membrane could drive the clustering of Bak proteins. This knowledge could ultimately enable us to learn how to manipulate apoptosis in cells, potentially as part of treatments for the diseases in which this cell death process occurs inappropriately.

How symmetric homodimers of Bak or Bax then associate to porate the mitochondrial outer membrane is still unknown. Although there is structural information for Bak and Bax homodimers, no structures of higher order oligomers of Bak or Bax have been resolved. Upon an apoptotic stimulus, Bak and Bax have been shown to coalesce into clusters at the mitochondria (*Nechushtan et al., 2001*) and small clusters of Bax have been implicated in the rapid release of cytochrome *c* (*Zhou and Chang, 2008*). The availability of full-length recombinant Bax makes biochemical assays more amenable for Bax than for Bak. Thus, recent studies using artificial membranes and recombinant Bax have detected Bax complexes of various shapes and sizes with fluorescence microscopy (*Subburaj et al., 2015*) and cryo-electron microscopy has revealed Bax protein exclusively associated with pore edges (*Kuwana et al., 2016*). Furthermore, super-resolution microscopy has shown Bax can be found in ring-like structures, arcs and clusters at the mitochondrial outer membrane of apoptotic cells (*Große et al., 2016*; *Salvador-Gallego et al., 2016*), although the orientation of Bax dimers in these structures could not be visualised. Higher order oligomers of both Bak and Bax vary in size when assessed by gel filtration, blue native PAGE or linkage (*George et al., 2007*; *Dewson et al., 2012*). Moreover, linkage studies have identified several points at which dimers might associate, including interactions at α-helices 1, 3, 5, 6, and 9 (*Dewson et al., 2008*, *2009*; *Zhang et al., 2010*; *Pang et al., 2012*; *Ma et al., 2013*; *Aluvila et al., 2014*; *Bleicken et al., 2014*; *Gahl et al., 2014*; *Iyer et al., 2015*; *Mandal et al., 2016*; *Zhang et al., 2016*). Critically, it is not clear whether any of these interaction sites are required for dimer-dimer association and assembly of the apoptotic pore. Here, we compare linkages through the full length of Bak in cells and show that dimers associate in a disordered and lipid-mediated fashion.

## Results

### The Bak N-segment, α1 and α1- α2 loop are solvent-exposed in Bak oligomers

To resolve how Bak dimers coalesce to porate the mitochondrial outer membrane, we sought to generate a more detailed biochemical map of the membrane topology of Bak dimers. This work complemented our previous cysteine-accessibility analyses of Bak α5, α6 and α9 (*Westphal et al.,*

*2014*; *Iyer et al., 2015*), by analyzing the Bak N-terminus and additional residues in the α2-α5 core and C-terminus. The two native cysteine residues of human Bak (C14 and C166) were first substituted with serine to generate Bak Cys null (BakΔCys, i.e. C14S/C166S), and then a single cysteine residue substituted at several positions throughout the molecule. Each Bak variant was stably expressed in *Bak⁻/⁻Bax⁻/⁻* mouse embryonic fibroblasts (MEFs) and tested for function (*Figure 1—figure supplement 1* and *Figure 1—figure supplement 2*). To convert Bak to the activated oligomeric form, membrane fractions enriched for mitochondria were incubated with tBid, as previously (*Dewson et al., 2008*). To label solvent-exposed cysteine residues, membrane fractions were treated with the thiol-specific labelling reagent IASD (4-acetamido-4'-((iodoacetyl) amino)stilbene-2,2'-disulfonic acid). Two negative sulfonate charges prevent IASD from accessing cysteine in hydrophobic environments (such as the mitochondrial outer membrane or the hydrophobic protein core), and also allow isoelectric focusing (IEF) to resolve IASD-labelled and IASD-unlabelled Bak (*Tran et al., 2013*; *Westphal et al., 2014*). We assessed each cysteine-substituted Bak variant for labelling before, during or after incubation with recombinant tBid or tBid^Bax, a tBid variant containing the Bax BH3 domain (*Hockings et al., 2015*), that activates Bak analogous to tBid, as well as a control not exposed to IASD and another fully exposed to labelling by denaturation and membrane solubilisation (*Figure 1A*, *Figure 1—figure supplement 3* and *Figure 1—source data 1*). The approach thus monitors solvent-exposure of the residue as non-activated Bak converts to oligomeric Bak, and may detect transient exposure during these conformational changes.

In the N-terminus, most tested residues were accessible to IASD even before tBid treatment (*Figure 1B*), consistent with the crystal structure (2IMT) (*Moldoveanu et al., 2006*). Those residues remained exposed following incubation with tBid. Four residues in α1 (R36C, Y41C, Q44C and Q47C) that were not fully accessible before tBid treatment became exposed following incubation with tBid (*Figure 1B*). An additional α1 residue, V39C, also showed a tendency for increased labelling. These changes suggest that the α1-α2 loop separates from α1, consistent with increased labelling of A54C in the α1-α2 loop that opposes α1. Thus, the increased labelling of cysteine residues placed throughout the N-segment, α1 and the α1-α2 loop, together with exposure of several N-terminal antibody epitopes (*Alsop et al., 2015*), indicate that the Bak N-terminus becomes completely solvent-exposed after activation (*Figure 1D*).

To complete our survey of the core and C-termini of Bak dimers (*Westphal et al., 2014*; *Iyer et al., 2015*), we tested the cysteine accessibility of additional selected residues in the α2-α5 core and the α7-α8 region (*Figure 1B*). A summary of current and previous cysteine labelling data for the full length of the Bak protein is shown in *Figure 1C* and supports the following picture of Bak dimer topology (*Figure 1D*). Certain cysteine residues placed in α2 became transiently exposed, consistent with exposure of the BH3 domain followed by its burial in the hydrophobic groove in the α2-α5 core dimer (*Figure 1C*), as predicted by the published structure of the Bak BH3:groove homodimer (4U2V) (*Brouwer et al., 2014*). Cysteine substitutions on the exposed surface of α3 were successfully labelled with IASD before, during and after tBid treatment. In contrast, two buried residues in α4 were refractory to IASD labelling throughout. In the α4-α5 linker region, although I123C became more exposed following activation, it was still incompletely labelled. This pattern of I123C labelling may reflect initial loss of contact between I123C and two hydrophobic residues in the 'latch' (I167 and W170), followed by shallow membrane insertion of I123C, as it resides on the hydrophobic face of the Bak dimer core. Partial labelling of this residue is consistent with shallow membrane insertion as IASD can label cysteine moieties up to 7.5 Å into the hydrocarbon core of a lipid bilayer (*Gründling et al., 2000*; *Westphal et al., 2014*). Within the C-termini, the profile of IASD-labelling was consistent with the amphipathic α6 helix lying in-plane on the membrane surface (*Westphal et al., 2014*) and the hydrophobic α9 forming the sole transmembrane domain (*Figure 1D*).

In summary, Bak undergoes a series of conformational changes as it transitions from its non-activated monomeric state, to an activated BH3-exposed monomeric intermediate, to the symmetric dimer that is the building block of the Bak oligomer (*Figure 1D*). In the course of these structural rearrangements, some regions of Bak display changes in solvent exposure (such as α1 which becomes more exposed, or α2 which is transiently exposed) whilst others remain buried throughout (such as the α9 transmembrane anchor).

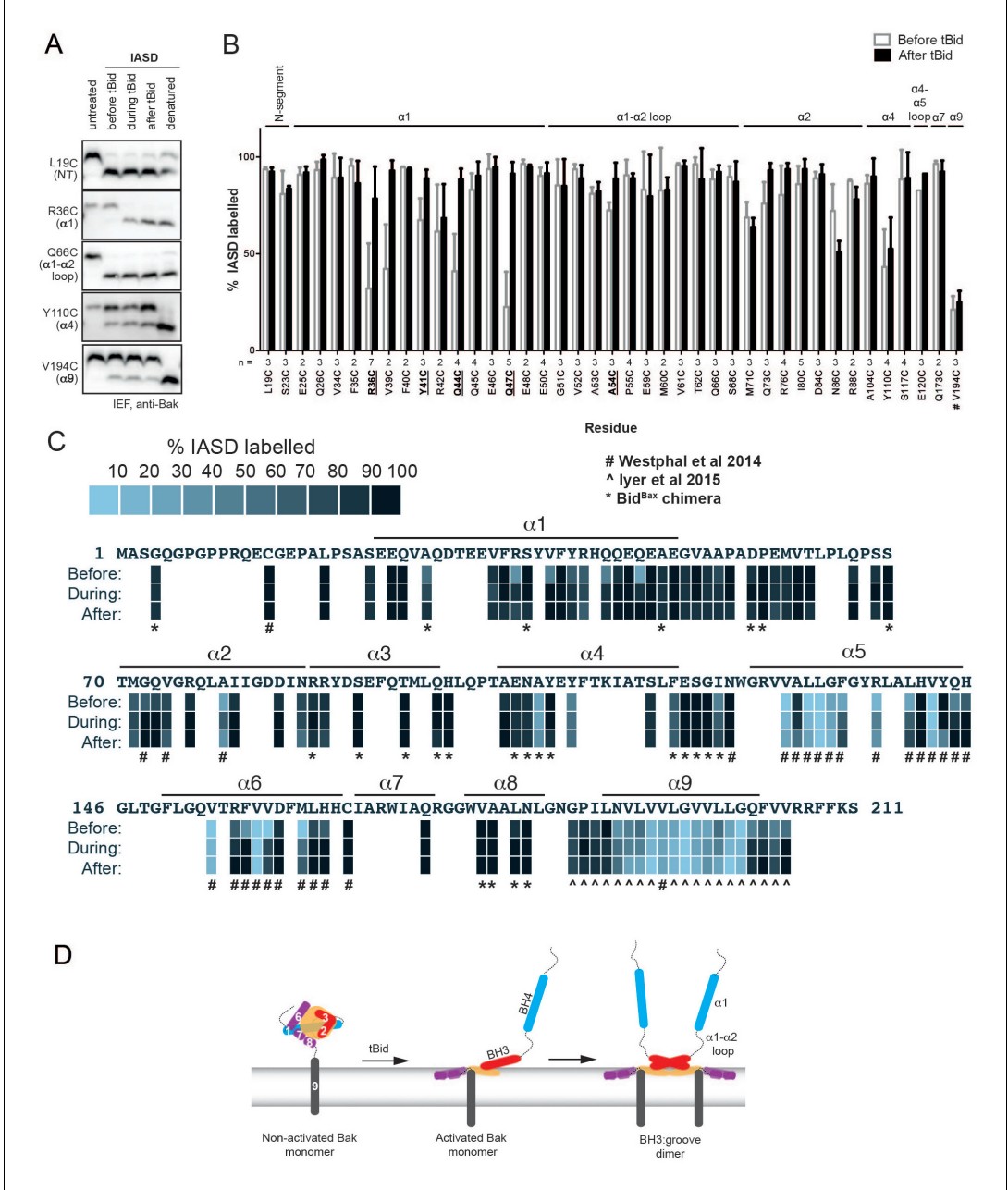

**Figure 1.** Following oligomerisation, the Bak N-segment, α1 and α1-α2 loop become fully solvent-exposed in contrast to the partially exposed core (α2-α5) and latch (α6-α9). (**A**) Solvent exposure of hBak cysteine mutants was assessed by IASD labelling before (lane 2), during (lane 3) and after (lane 4) treatment with tBid. Controls of unlabelled (untreated, lane 1) and fully labelled (denatured, lane 5) Bak were included for comparison. Example IEF western blots are shown. (**B**) Quantitation of IASD labelling before and after treatment with tBid for the panel of previously untested Bak residues. Data are mean ± SD (n ≥ 3), or range (n = 2), with n for each residue labelled on the x-axis. IASD labelling data for residue V194C were from (**Westphal et al., 2014**) (denoted #). Residues for which there is a significant difference in IASD labelling before versus after tBid are in bold and underlined (p<0.05). (**C**) Heat map overview of Bak IASD labelling with tBid treatment from (**B**) pooled with published analyses from (**Westphal et al., 2014**)(denoted #) or (**Iyer et al., 2015**)(denoted ˆ) and from treatment with the tBid^Bax chimera (denoted *; see also **Figure 1—figure supplement 3**; **Figure 1—source data 1**). (**D**) Schematic of Bak structural rearrangement from its non-activated monomeric state to the activated dimer. Helices are numbered for the non-activated Bak. Note the complete solvent-exposure of α1 and the α1-α2 loop in the activated dimer. The following figure supplements are available for **Figure 1**.

The following source data and figure supplements are available for figure 1:

**Source data 1.** Quantitation of Bak IASD labelling before, during and after Bak activation; table of values.

*Figure 1 continued on next page*

*Figure 1 continued*

**Figure supplement 1.** Bak cysteine variants retain apoptotic function.

**Figure supplement 2.** Bak cysteine variants retain apoptotic function in response to tBid.

**Figure supplement 3.** Quantitation of Bak IASD labelling before, during and after Bak activation; graphical output.

## The N- and C-termini of activated Bak are mobile

Linkage studies have provided significant insight into structural changes in Bak, as well as how the activated proteins associate into higher order oligomers (*Dewson et al., 2008*, *2009*; *Ma et al., 2013*; *Aluvila et al., 2014*; *Brouwer et al., 2014*; *Iyer et al., 2015*; *Mandal et al., 2016*). To directly compare reported linkages, and to analyse the full length of Bak, we used our expanded library of single-cysteine Bak variants. Each substituted cysteine was tested for the ability to disulphide bond to the same residue in a neighbouring Bak molecule (or to a cysteine in a nearby protein) upon addition of the oxidant copper phenanthroline (CuPhe). Linkage between Bak molecules was indicated by the presence of 2x complexes corresponding to twice the molecular weight of the Bak monomer on non-reducing SDS-PAGE. It is important to note that these 2x complexes are the product of CuPhe-mediated linkage between Bak molecules and do not necessarily represent the native Bak BH3:groove dimer that will form even in the absence of linkage, as shown below on BNP. This system of single cysteine substitutions offers an elegant screening approach for assessing the proximity of single cysteine residues in neighbouring molecules in a point-to-point manner, but does not identify interaction surfaces where a single cysteine substitution may not contact its counterpart (See Discussion).

Even in the absence of tBid treatment, cysteine residues in flexible regions displayed some linkage to neighbouring non-activated Bak molecules (*Figure 2*, upper panels). For example, some 2x complexes were captured by cysteine substitutions in the flexible N-segment (G4C, C14, L19C, S23C), and α1-α2 loop (G51C, D57C, P58C, Q66C, S68C, S69C), indicating some proximity of Bak monomers in untreated mitochondria. Furthermore, disulphide linkage to proteins other than Bak resulted in Bak complexes larger than the 2x species and was observed for cysteine substitutions in the α2-α3 helices (D84C, R87C), α4-α5 loop (S121C, N124C), α8-α9 linker (G186C, I188C) and C-terminal end of α9 (V205C). This is unsurprising as the MOM is a protein-rich environment and there are documented examples of monomeric Bak associating with other membrane proteins such as VDAC2 (*Cheng et al., 2003*; *Lazarou et al., 2010*; *Ma et al., 2014*).

After addition of tBid had activated Bak, each cysteine in the N-extremity (a region comprising the N-terminus, α1 and α1-α2 loop) showed more efficient linkage to 2x complexes (*Figure 2A*, upper panel). The freedom with which each cysteine in the N-extremity of Bak can link suggests this region is highly mobile and does not engage in stable protein-protein or protein-lipid interactions following activation. In contrast, cysteine residues positioned in the core (α2-α5) showed relatively modest linkage to 2x complexes (*Figure 2B*, upper panel), consistent with lack of flexibility in the α2-α5 core dimer (*Brouwer et al., 2014*) and its relative immobility due to association with the membrane.

We next addressed linkage in the C-extremity encompassing the α6-α8 latch and α9 transmembrane domain (*Figure 2C*, upper panel). After Bak activation, H164C in α6 showed increased linkage to 2x complexes, as reported previously for several residues in α6 (*Dewson et al., 2009*; *Ma et al., 2013*). Cysteine substitutions in the α8-α9 loop and the beginning of the α9 transmembrane domain (up to residue N190C) also showed linkage to 2x complexes (*Figure 2C*, upper panel). However, compared to the N-extremity, linkage was less complete, possibly due to membrane association limiting mobility. Linkage of membrane-buried α9 residues was even less efficient (*Figure 2C*). However, at a higher temperature α9 residues show much stronger linkage (*Figure 2—figure supplement 1*) (*Iyer et al., 2015*), attributable to increased Bak mobility or CuPhe penetration into the bilayer.

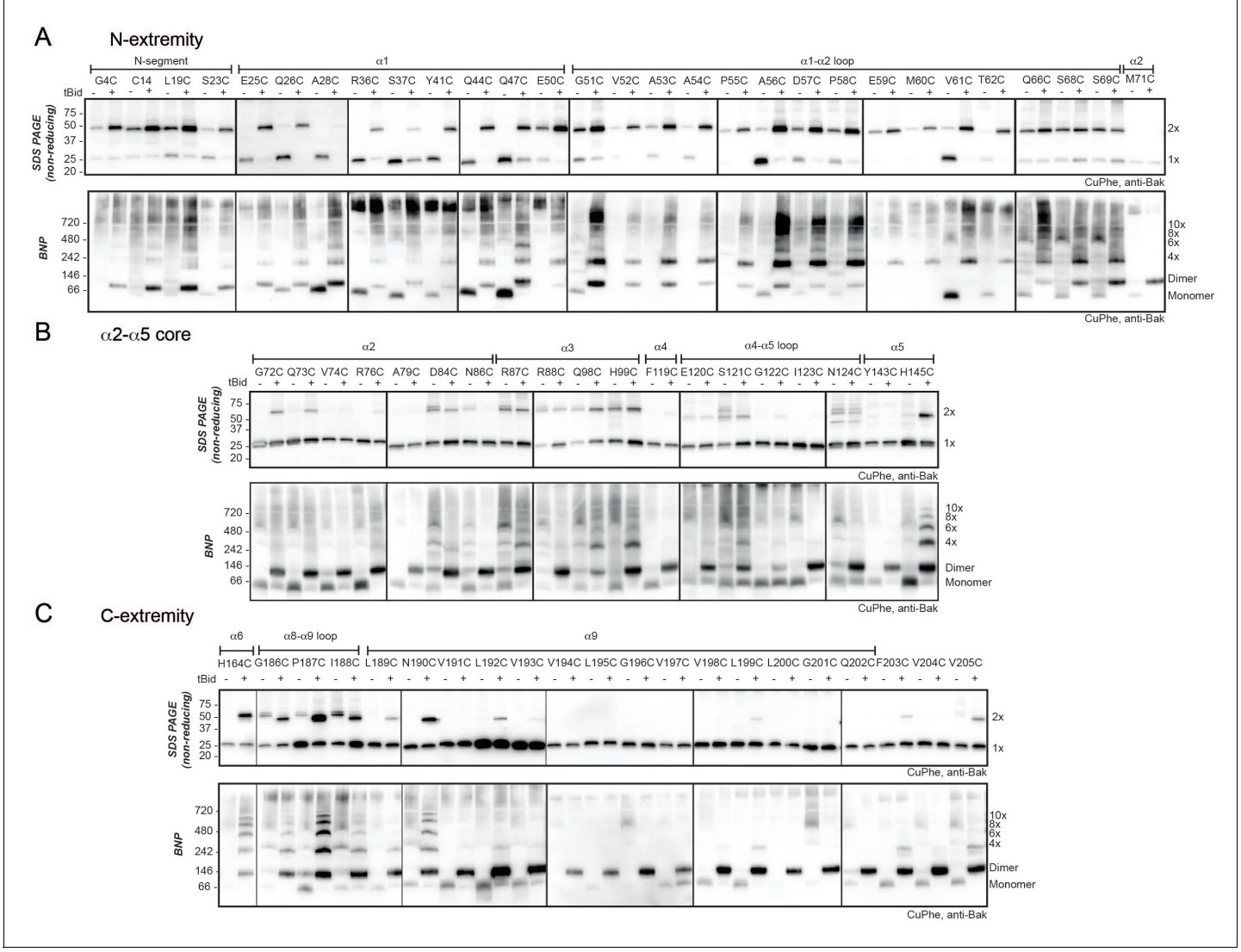

**Figure 2.** The N- and C-extremities of oligomerised Bak are mobile relative to the α2-α5 core dimer. Mitochondrial fractions from cells expressing the indicated Bak cysteine mutants were incubated with tBid to oligomerise Bak, and oxidant (CuPhe) added to induce disulphide bonds. Aliquots were analyzed by non-reducing SDS PAGE (upper panels) and BNP (lower panels), and immunoblotted for Bak to detect linked species. Data are representative of at least two biological replicates. (**A**) Cysteine linkage occurs throughout the N-extremity. (**B**) Cysteine linkage occurs, but is less complete, in the α2-α5 dimer core. (**C**) Cysteine linkage occurs in the C-extremity. (See also α6:α6 linkage in *Dewson et al. [2008]* and α9:α9 linkage in *Iyer et al. [2015]*).

The following figure supplement is available for figure 2:

**Figure supplement 1.** Higher temperature enhances disulphide bonding of cysteine residues in α6 (H164C) and the α9 transmembrane domain (L199C).

In summary, after Bak activation, the whole N-extremity is highly dynamic due to being fully solvent-exposed; the core is constrained and membrane-associated; and the C-extremity is flexible, but is anchored to the membrane.

## Mobility of the N- and C-extremities permits linkage between dimers

We next used blue native PAGE (BNP) to consider two types of cysteine-mediated Bak linkage: linkage *within* Bak dimers and linkage *between* Bak dimers. The Bak BH3:groove interaction *within* dimers is an extensive protein-protein interface and is stable in 1% digitonin on BNP while the

interaction between dimers is not (*Ma et al., 2013*). Thus oligomerised Bak migrates only as BH3: groove dimers on BNP. Addition of CuPhe to mitochondrial extracts prior to BNP can stabilise high order complexes on BNP and does so by cysteine-mediated linkage *between* dimers (*Ma et al., 2013*). Thus, if the addition of CuPhe at the end of the mitochondrial incubation generated 4x and greater complexes on BNP, we conclude that the linkage was *between* dimers. If larger complexes were absent and dimers predominated on BNP, any linkage was *within* the dimer. Thus, parallel examination of CuPhe linked Bak species on BNP and non-reducing SDS PAGE permits differentia- tion of three conditions: the linkage *within* the dimer, linkage *between* dimers, or no cysteine linkage (*Figure 3A*).

In the N-extremity (G4C to S69C), each introduced cysteine could link some dimers to larger com- plexes on BNP (*Figure 2A*, lower panel). Upon tBid activation, SDS-PAGE (*Figure 2A*, upper panel) showed an increased proportion of 2x complexes, and higher order complexes (>10x) were observed on BNP for all residues. Of note was a stretch of residues from approximately E59 to T62 that, following tBid treatment, exhibited very efficient linkage by SDS-PAGE (obvious 2x, negligible 1x). In this stretch, BNP showed no dimer species and a concomitant shift to 4x species and higher order complexes, indicating efficient linkage between dimers but no linkage within dimers. Residues closer to the α2-α5 core (Q66C to S69C), showed progressively more 1x species (SDS-PAGE) and an increase in dimers (BNP), indicative of becoming closer to the constrained core of the dimer. Together, these linkage data further argue that dimers possess flexible N-extremities capable of effi- cient linkage within and between dimers, with linkage becoming exclusively between dimers as resi- dues approach the constrained α2-α5 core (*Figure 3B*). Interestingly, because the linkage of cysteine residues in the E59-T62 region occurs *only* between dimers, and is very efficient, this linkage provides a molecular sensor that can specifically quantitate dimer-dimer interactions, and is the first assay to do so.

The V61C:V61C' molecular sensor for dimer-dimer interaction is generalisable to other stimuli and to the Bax pore (*Figure 3—figure supplement 1*). For example, the same V61C:V61C' crosslink- ing pattern for Bak is induced by either tBid or heat treatment, arguing that the linkage pattern between dimers is a general characteristic of the Bak apoptotic pore. We also generated the V61C equivalent in membrane localised Bax (Bax A46C/S184L) (*Figure 3—figure supplement 1*). This Bax variant could be linked between (but not within) dimers, suggesting that Bax dimers also have flexi- ble N-terminal extremities.

In the Bak C-terminus also, cysteines could link dimers to larger complexes on BNP, although a striking laddering pattern was apparent (*Figure 2C*, lower panels). Absence of the large (>10x) com- plexes common at the N-extremity is explained by the relatively restricted range of motion of the membrane-associated C-extremity. The very weak laddering pattern for certain residues in the α6-α 8 region is explained by those residues facing either the membrane or the cytosol rather than lat- erally to a nearby Bak dimer. Lastly, linkage in the C-extremity may be only between dimers (rather than within dimers) as the proportion of 2x to 1x species on SDS-PAGE is similar to the proportion of higher order complexes to dimers on BNP.

## Single-cysteine mutants in the rigid core can capture some linkage, mostly at the lateral edges

Previous studies suggested that the structured α2-α5 core of the Bak (or Bax) dimer might form a dominant, symmetric interface capable of driving dimers to form higher order complexes (*Aluvila et al., 2014*; *Brouwer et al., 2014*). For example, crystals of the Bak α2-α5 core dimer showed a side-by-side association (4U2V)(*Brouwer et al., 2014*). However, this was attributed to crystal packing effects, as a chemical crosslinking between the α4-α5 loops was not detected (*Brouwer et al., 2014*). Accordingly, in the present studies, disulfide bonding between the α4-α5 loops was not captured with CuPhe (*Figure 2B*, lower panel). End-to-end associations were sug- gested by α3:α3' or α5:α5' linkages (*Aluvila et al., 2014*; *Brouwer et al., 2014*; *Mandal et al., 2016*), and disulfide bonding at the equivalent residues was also evident in this study (H99C:H99C for α3:α3'; H145C:H145C' for α5:α5')(*Figure 2B*, lower panel). Notably, this linkage was not as effi- cient as linkage observed for the N-extremity, and these and other linkage-competent residues locate mostly to the lateral edges of the α2-α5 core dimer when lying in-plane (*Figure 3C*). Thus, screening individual cysteine substitutions in the Bak dimer core for linkage did not uncover any

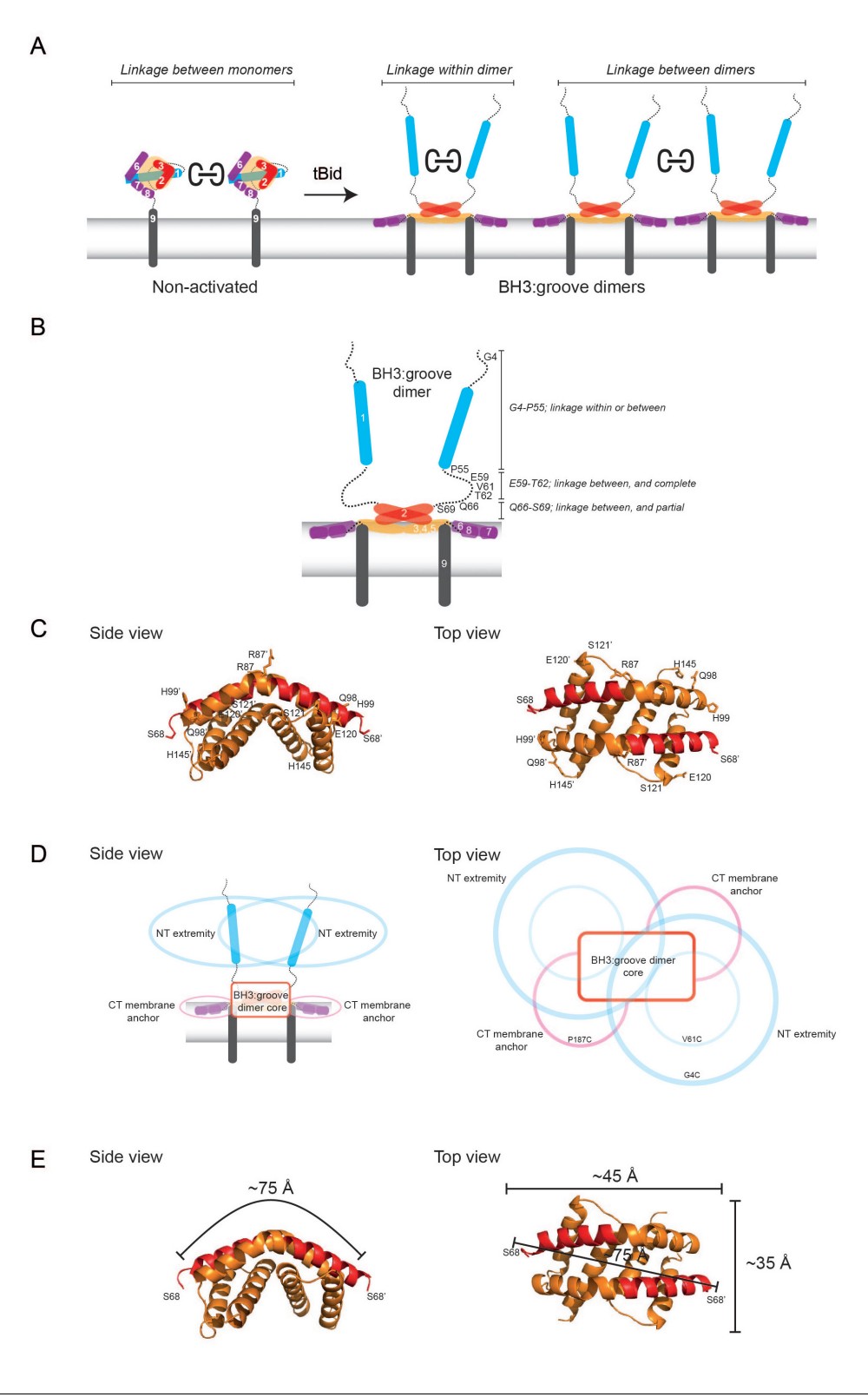

**Figure 3.** Linkage constraints for different regions of the active Bak dimer support a flexible extremity model for full-length Bak dimers in the MOM. (**A**) The oxidant CuPhe induces linkage *between* Bak monomers, and *within* or *between* Bak dimers. Correlation of BNP with non-reducing SDS PAGE can differentiate linkage *within* and *between* dimers. (**B**) Summary of linkage outcomes *within* or *between* dimers for residues in the Bak dimer N-extremity. (**C**) Linkage at the lateral corners of the Bak α2-α5 core dimer is highlighted with residue labels and sticks on the crystal structure (4U2V)

*Figure 3 continued on next page*

*Figure 3 continued*

[*Brouwer et al., 2014*]). (D) Schematic of a dimer of activated Bak at the membrane in side and top views. The range of movement of the N- and C-extremities is indicated with coloured ovals (blue and pink respectively). (E) Our linkage data are consistent with the dimensions of the Bak core dimer. The dimensions of the Bak α2-α5 core dimer (4U2V (*Brouwer et al., 2014*) are shown in side view and top view. In the crystal structure, the distance from one edge of the core at S68 over the bended structure of the symmetric core dimer to the opposing S68' is ~75 Å. Thus, for residues in the mobile N-extremity to link within the core dimer, the two homotypic cysteine residues must bridge a distance of ~75 Å. This is not feasible for the residue V61C which is ~30 Å from the symmetric core.

The following figure supplement is available for figure 3:

**Figure supplement 1.** Bak V61C linkage is generalisable to a heat stimulus and to the linkage of the equivalent Bax A46C residue.

dominant sites of interaction between Bak dimers, but showed widespread linkage of residues around the lateral edges of the dimer, consistent with the random collision of Bak dimers.

These findings support a flexible extremity model for full-length Bak dimers in the MOM, outlined in *Figure 3D* with side-on (left) and top-down views (right). At the centre is the α2-α5 core dimer lying in-plane and partially submerged in the membrane surface. Extending from both sides of the core dimer is a flexible latch (α6-α8) of ~40 residues that also lies in-plane and connects to the trans-membrane α9-helices. Extending from both ends of the dimer core is the solvent-exposed, flexible and mobile N-extremity of ~70 residues. Each circle in *Figure 3D* illustrates the range of possible positions for the flexible N- and C-extremities (blue and pink respectively). Overlap of the largest blue circles (labelled G4C) indicates the potential for residues close to the N-terminus to link within dimers. In contrast, the smaller blue circles (labelled V61C) do not overlap, indicating a lack of link-age within the dimer at this residue. Notably, these linkage constraints are consistent with the dimensions of the α2-α5 dimer core as determined from the recent crystal structure (4U2V) (*Figure 3E*).

These findings also reveal how dimers associate into high order oligomers. As linkage can be induced between several regions of the Bak dimer, there is no dominant protein-protein interface that mediates dimer assembly into high order oligomers. Rather, the linkage pattern is consistent with the transient collision of dimers in-plane on the membrane. Collisions of the α2-α5 core and α6-α8 latch are limited by their membrane-association, whereas the flexible, entirely solvent-exposed N-terminal region is free to link between dimers and does so very efficiently.

## Inter-dimer interfaces are more labile to detergent than the intra-dimer (BH3:groove) interface

We next examined if the inter-dimer interactions identified above were stable in digitonin, as shown for the BH3:groove interface within dimers (*Ma et al., 2013*). Digitonin was added to the mitochondrial incubations after Bak had become oligomerised by incubation with tBid, but prior to disulphide bond formation induced by CuPhe (*Figure 4A*). As expected, digitonin did not prevent linkage *within* dimers as shown by 2x complexes of M71C/K113C on SDS-PAGE (*Figure 4A*, upper panel, lane 3). However, digitonin prevented all linkage *between* dimers on BNP of seven single-cysteine variants (*Figure 4A*, lower panels), indicating that inter-dimer interactions are mediated either by membrane or by weak protein-protein interactions.

These experiments also provided new insight into linkage within dimers. For example, two residues at the far N- and C-termini (L19C and the C-terminal extension GGSGGCK [*Iyer et al., 2015*]) could link within dimers to generate 2x complexes on SDS-PAGE after digitonin treatment (*Figure 4A*, upper panels). Thus, these residues were sufficiently distal from the core to reach their counterpart across the length of the structured α2-α5 core dimer and allow linkage within the dimer (*Figure 3D and E*). On BNP, dimers of these two residues (L19C, and to a lesser extent GGSGGCK) also migrated faster than dimers of other variants (*Figure 4A*, lower panels), suggesting that linkage within the dimer caused a more compact, faster migrating protein complex under these native conditions.

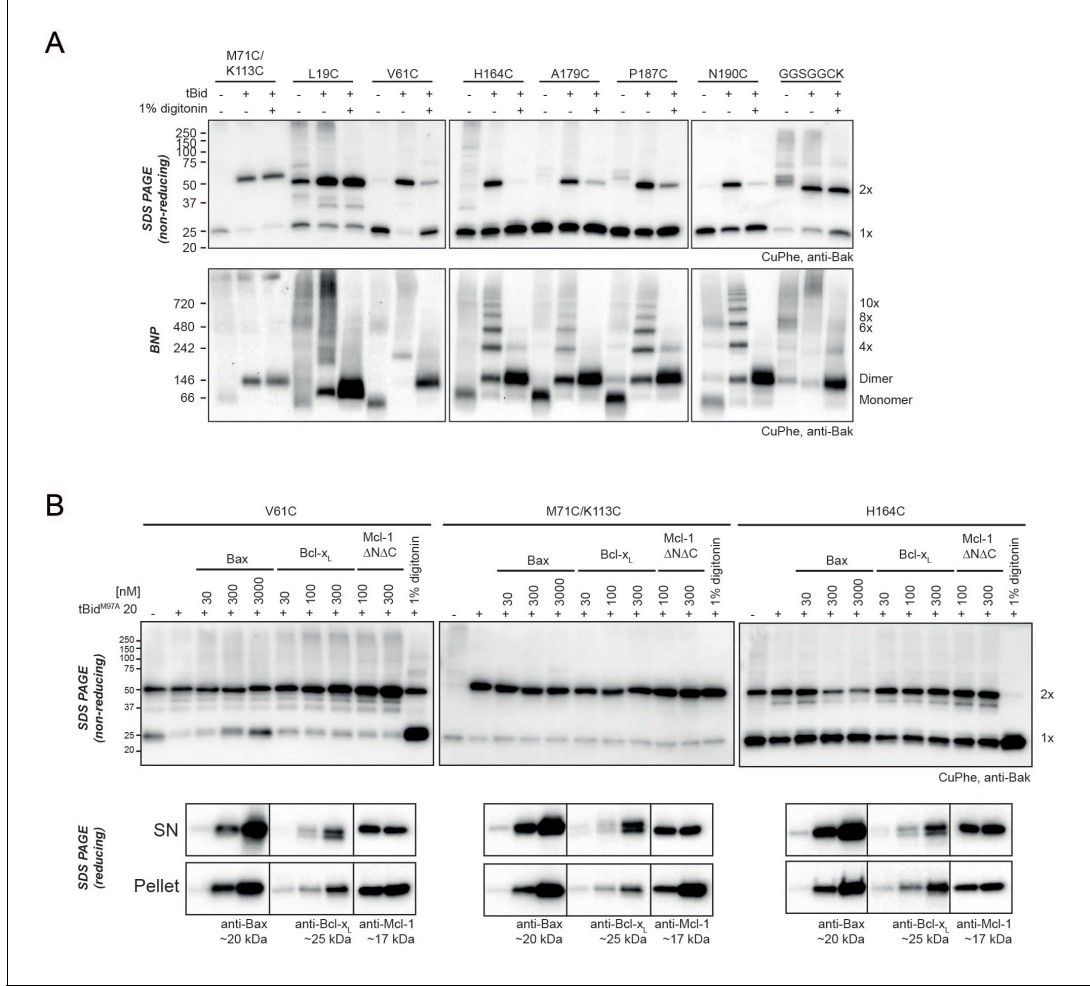

**Figure 4.** Bak dimer-dimer interactions are disrupted by detergent or Bax. (**A**) Digitonin prevents linkage between Bak dimers. Membrane fractions expressing the indicated Bak single-cysteine variants were first treated with tBid, and then supplemented, as indicated, with detergent (1% digitonin) prior to cysteine linkage. Two additional mutants were included to show linkage within dimers at the BH3:groove interface (M71C/K113C) and at extensions to the C-terminus (GGSGGCK). Data are representative of two biological replicates. (**B**) Bax can disrupt Bak dimer-dimer association. Membrane fractions first treated with tBid$^{M97A}$ were then incubated, as indicated, with Bax, Bcl-x$_L$, Mcl-1ΔN151ΔC23 or digitonin prior to cysteine linkage (upper panels). Supernatant and pellet fractions showed partial localisation of each recombinant protein to the membrane fraction (lower panels). Data are representative of three biological replicates.

## Bak dimers can become interspersed by Bax

To test if interactions between Bak dimers could be disrupted by other proteins, and thus test if Bak dimer aggregation was a dynamic, reversible process, we added recombinant mitochondrial proteins after Bak had formed dimers (*Figure 4B*). Dimer interactions were monitored by linkage between the N-termini (V61C:V61C') or C-termini (H164C:H164C'), and compared to BH3:groove linkage (M71C:K113C). To oligomerise Bak, mitochondria were first incubated with tBid$^{M97A}$, a variant that binds poorly to prosurvival proteins such as Bcl-x$_L$ and Mcl-1 (*Lee et al., 2016*). Mitochondria were then incubated with the recombinant Bcl-2 proteins Bax, Bcl-x$_L$ or Mcl-1ΔNΔC. Finally, CuPhe was applied and the extent of disulphide linkage assessed by non-reducing SDS-PAGE (*Figure 4B*). Notably, Bax localised to mitochondria and decreased linkage between Bak dimers (e.g. V61C: V61C' and of H164C:H164C') but not within dimers (e.g. M71C:K113C) (*Figure 4B*). In contrast, recombinant Bcl-x$_L$ and Mcl-1ΔNΔC localised to the mitochondrial membrane but did not interfere with the linkage between Bak dimers (*Figure 4B*). Thus, Bax not only localised to the same

membrane microdomain as the pre-formed Bak oligomers, but became partially interspersed with the dimers, indicating that dimer aggregation is dynamic.

## V61C:V61C' linkage can monitor dimer-dimer interactions; stochastic simulations reveal a disordered aggregate of Bak dimers

Our biochemical data support the random collision of Bak dimers at the mitochondrial outer membrane during apoptosis. Indeed, several studies indicate that Bak and Bax can reside in a variety of different structures at the mitochondrial outer membrane of apoptotic cells, including clusters (*Nechushtan et al., 2001*; *Zhou and Chang, 2008*; *Große et al., 2016*; *Nasu et al., 2016*; *Salvador-Gallego et al., 2016*). To test if random collisions between dimers could feasibly explain the linkage patterns observed, simulations of the random contact between dimers were performed. The efficient and exclusive linkage that occurs between dimers at residue V61C formed the basis of our simulations (*Figure 5*).

To simulate the aggregation of Bak dimers on the mitochondrial outer membrane, this complex biological system was simplified to limit the number of parameters required to mimic the experimental outcome. We simulated the membrane as a flat surface (i.e. a 2D plane). Each subunit representing a Bak dimer, was evenly distributed on a geometric grid imposed on this surface (*Figure 6A*), and to avoid edge effects the grid was wrapped onto a torus. A hexagonal grid, rather than square or triangular, was selected to best represent the close packing of objects on a plane with the greatest degrees of freedom. The linkage potential of each subunit in the grid was based on the known linkage constraints of the V61C residue; i.e. linkage occurred exclusively and efficiently between dimers and each dimer could make at most two cysteine linkages, but possibly only 1 or 0 linkages if no other free cysteines were available. Furthermore, the simulation allowed linkages only between immediately adjacent dimers, based on the short 4 Å limit of the CuPhe-mediated disulphide linkage and the known dimensions of the Bak dimer. Thus, each hexagonal subunit had the freedom to link with at most two neighbouring hexagons, in any direction, and this random linkage between neighbouring dimers would result in a mixture of linkage states including double, single or no linkages.

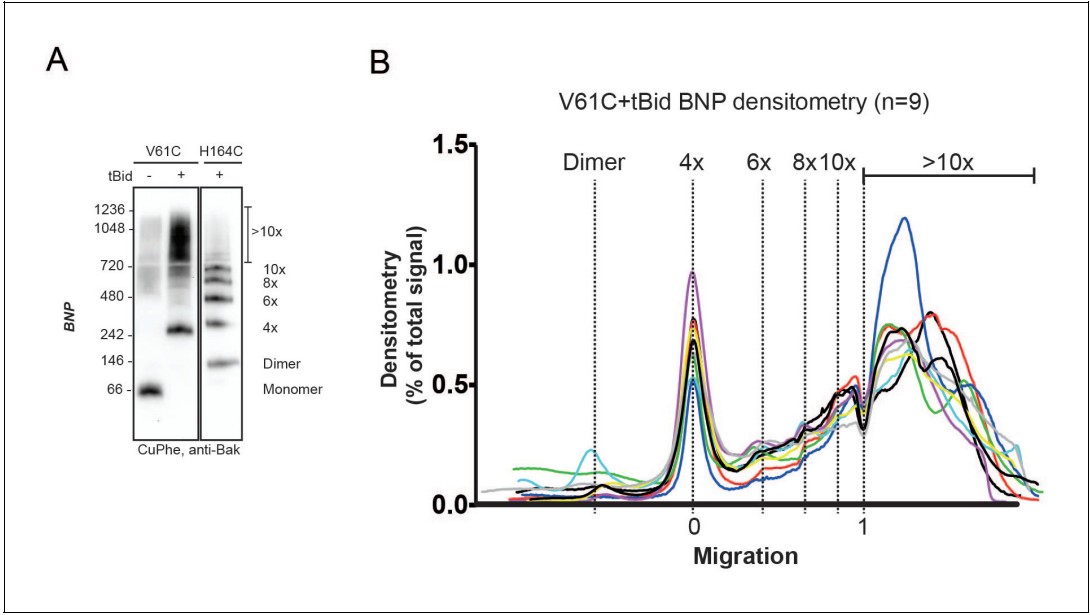

**Figure 5.** V61C:V61C' linkage is a marker of dimer-dimer interaction. (**A**) BNP analysis of V61C:V61C' linkage revealed no dimers, and a high proportion of 4x and >10x species. In contrast, H164C:H164C' linkage revealed a ladder of linked species. Data are representative of nine biological replicates. (**B**) Densitometry of V61C:V61C' BNP linkage in Bak oligomers shows negligible dimers but reproducible linked species at 4x, 6x, 8x, and 10x, and a large population of species > 10x. Densitometry data (n = 9) were normalised to the area under the curve after alignment as described in Materials and Methods. To correct for small variations in electrophoretic migration, we employed a noise reduction algorithm in which the 4x peak (0) and a minima at ~720 kDa (1) were aligned between replicates.

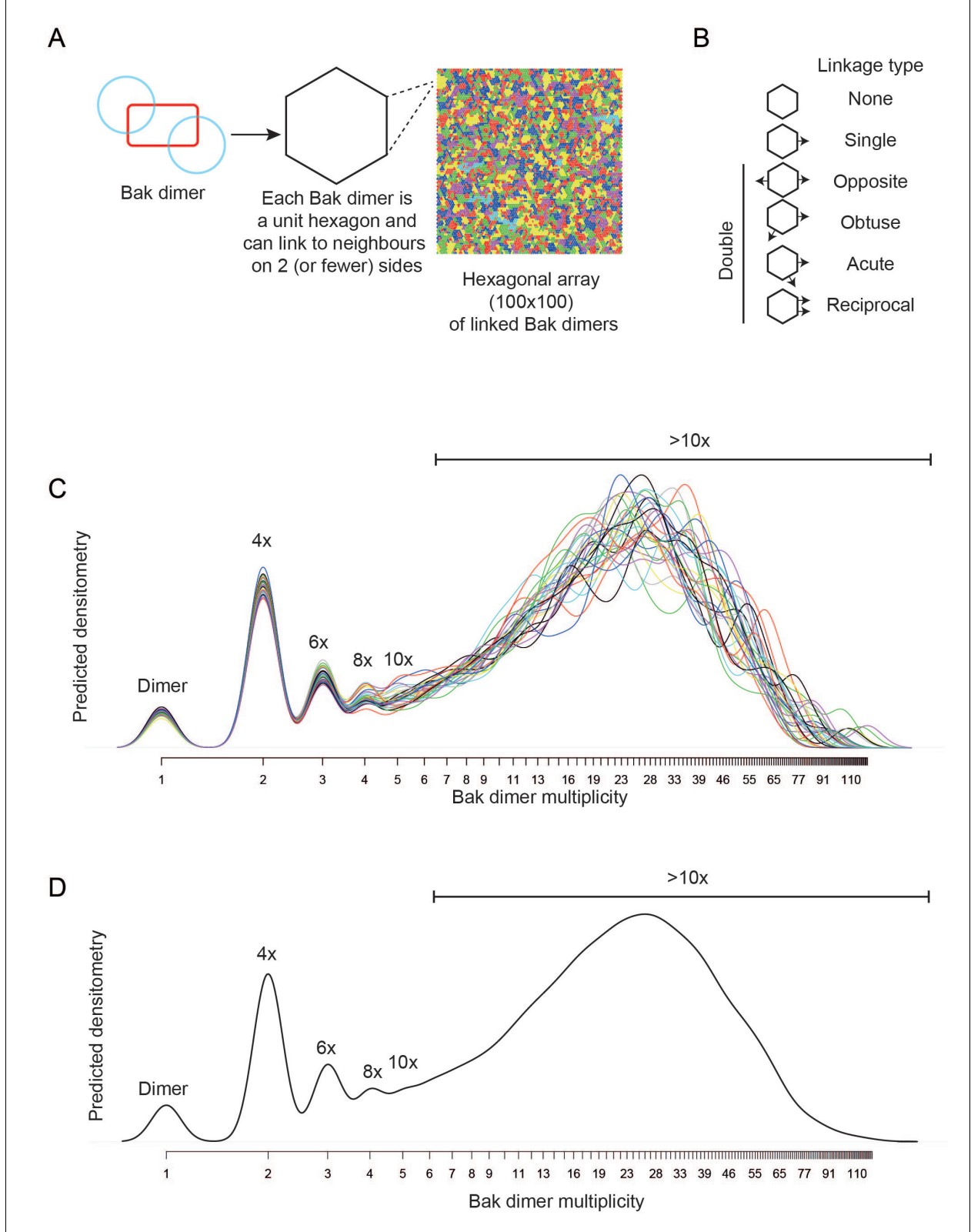

**Figure 6.** Bak dimer arrangement examined by two-dimensional stochastic simulations: a random arrangement successfully models the V61C:V61C' BNP linkage densitometry. (**A**) A grid of 100 × 100 was designated for the 2D simulations. Each unit hexagon within that grid represented a Bak dimer, with the capacity to link to two neighbouring hexagons. The direction of linkage from each hexagon was randomised. An example visual output from a

*Figure 6 continued on next page*

*Figure 6 continued*

100 × 100 hexagonal array is shown: neighbouring subsets of linked Bak dimers within the grid are delineated by different colours. (**B**) Linkage possibilities between each dimer unit in a hexagonal 2D array. (**C,D**) Overlay (**C**), and average (**D**) of 30 predicted densitometry plots.

The following figure supplements are available for figure 6:

**Figure supplement 1.** Adjustment to simulation output to allow qualitative comparison to western blot data.

**Figure supplement 2.** Comparison of 2D simulation using a square or hexagonal array reveals that either geometry offers a reasonable fit for the empirical data.

**Figure supplement 3.** 2D simulation with reduced efficiency linkage successfully models reduced linkage in mitochondrial experiments.

**Figure supplement 4.** 2D simulation with edge blocking probability successfully models mitochondrial experiments in which Bak linkage becomes constrained as cysteines are positioned closer to the dimer core.

With this hexagonal grid, four types of double linkage were possible; opposite, obtuse, acute or reciprocal (*Figure 6B*). An equal probability for each of the double linkages (opposite, obtuse, acute or reciprocal) was imposed, to reflect close but random packing of dimers at the membrane. Furthermore, the simulation was allowed to proceed to complete linkage of available cysteine residues to reflect the near-complete linkage of V61C:V61C'. With these few parameters, a complex biological system was elegantly reduced to a simplified simulation.

The simulation output was a frequency distribution of the number of Bak dimers present in each linked complex (*Figure 6—figure supplement 1*). To approximate the BNP western blot densitometry, the counts were then multiplied by the number of Bak dimer units (e.g. 1, 2, 3, 4 etc) in each of the linked Bak complexes, and the predicted densitometry smoothed with a Gaussian kernel to allow qualitative comparison with the Gaussian-like densitometry output from the BNP western blots.

Notably, the simulation produced outputs that closely matched the distinctive linkage pattern observed for V61C, i.e. prominent 4x complexes and complexes greater than 10x the Bak molecular weight (compare *Figure 5A and B* with *Figure 6C and D*). (When the simulation was repeated on a square rather than hexagonal grid, the output also closely matched the V61C linkage pattern, as shown in *Figure 6—figure supplement 2*). Thus, the simulations provided proof of principle that the linkage pattern (V61C:V61C') observed in mitochondria can be explained by random dimer arrangement. These logic arguments based on the linkage constraints of the V61C residue have, for the first time, afforded single molecule resolution of Bak aggregation.

## Simulations are robust to a reduction in linkage

To explore the robustness of the simulations, we asked if the simulation could predict the linkage pattern when linkage was limited due either to less V61C residues, or to cysteine residues at positions other than V61C. In the first test, we performed mitochondrial experiments in which ~50% of the Bak molecules lacked cysteine residues. In those experiments, Bak V61C was co-expressed with Bak Cys-null that was FLAG-tagged (*Figure 6—figure supplement 3A*, upper panel). As expected, only half of the Bak molecules could link to 2x species after activation by tBid and induction of linkage (*Figure 6—figure supplement 3A*, middle panel, lane 4). In addition, all molecules formed BH3: groove dimers after incubation with tBid, as shown by BNP, but the capture of higher order oligomers mediated by cysteine-linkage was greatly reduced (*Figure 6—figure supplement 3A*, lower panel, lane 4; quantified in *Figure 6—figure supplement 3B*). In simulations of this system, we assumed an equal expression of V61C and the Cys null variants, as indicated by western blot analysis. We also assumed that this population of monomers forms dimers in Mendelian proportions: 25% of dimers are V61C doublets and can link twice, 50% are V61C:Cys null and can link once, and the remaining 25% are Cys null doublets and cannot link (*Figure 6—figure supplement 3C*). The simulation output (*Figure 6—figure supplement 3D*) reflected that of the BNP densitometry (*Figure 6—figure supplement 3B*) in terms of more dimers and fewer >10x complexes, indicating the simulation was generalisable to a different distribution of linkage-competent Bak molecules.

We next examined if our simulations were also robust when approximating the linkage pattern of residues other than V61C (*Figure 6—figure supplement 4A*). As noted above, because T62C, Q66C, S68C, and S69C substitutions were closer than V61C to the constrained α2-α5 core, they did not link efficiently (*Figure 2A* lower, quantified in *Figure 6—figure supplement 4B*). To simulate this steric hindrance caused by a shorter distance from the constrained core, an *edge blocking* effect was introduced to the model; every edge of every hexagon was prohibited from having a link form across it with this probability for the duration of each simulation. As the edge blocking probability increased from 0 to 0.75, linkage between dimers decreased to generate patterns comparable to BNP analysis of these core-proximal residues (*Figure 6—figure supplement 4C*). Thus, modification of a single parameter could describe linkage of residues closer than V61C to the Bak dimer core. The adaptability of this simple simulation further illustrated the strength and feasibility of our model of random Bak dimer aggregation at the mitochondrial outer membrane during apoptosis.

## Discussion

Here we investigated the topology of Bak dimers in the mitochondrial outer membrane, and how dimers assemble into the high order oligomers thought necessary to form apoptotic pores in that membrane. We found that Bak dimers are characterised by flexible N- and C-extremities flanking a rigid α2-α5 core, and that these dimers aggregate in disordered, dynamic, clusters. Critically, we found no evidence for a single, dominant protein-protein interface between dimers and predict the lipid environment plays a crucial role in facilitating the aggregation of dimers and subsequent membrane rupture.

We had previously proposed the in-plane model for Bak dimers (*Westphal et al., 2014*). Our current data advances this model to show the N- and C-extremities are flexible, based on analysis of the full-length of Bak when activated and dimerised at the mitochondrial outer membrane (*Figure 3D*). The N-extremity (~70 residues N-terminal to α2) becomes fully solvent-exposed as shown by IASD labelling, and fully mobile as indicated by linkage, consistent with exposure of N-terminal cleavage sites and antibody epitopes in α1 and the α1-α2 loop (*Griffiths et al., 1999*; *Weber et al., 2013*; *Alsop et al., 2015*). As discussed previously (*Alsop et al., 2015*), several parts of the α1 region are able to unfold, as they bind antibodies that recognise linear epitopes. Failure of the N-extremity to re-engage in protein-protein or protein-lipid interactions argues that it does not contribute to the assembly of dimers. The remainder of Bak is membrane-associated, with the α2-α5 core and α6 (and possibly α7-α8) in-plane with the outer mitochondrial membrane surface and partially embedded (*Aluvila et al., 2014*; *Brouwer et al., 2014*; *Westphal et al., 2014*) while α9 forms a transmembrane domain (*Iyer et al., 2015*). The topology of the α2-α5 region is consistent with the α2-α5 core dimer crystal structure (~35 Å × 45 Å)(*Brouwer et al., 2014*), as V61C (~30 Å from the core dimer) linked between dimers but not within dimers. Flexibility of the C-terminal α6-α9 region, as indicated by multiple linkages between these regions (*Bleicken et al., 2014*; *Iyer et al., 2015*; *Zhang et al., 2016*), implies that embedded dimers adopt a range of conformations on the MOM surface (*Figure 7A*).

Linkage experiments have identified several possible interactions between dimers of Bak or Bax (*Dewson et al., 2008*, *2009*; *Zhang et al., 2010*; *Pang et al., 2012*; *Ma et al., 2013*; *Aluvila et al., 2014*; *Bleicken et al., 2014*; *Gahl et al., 2014*; *Iyer et al., 2015*; *Mandal et al., 2016*; *Zhang et al., 2016*), with many also shown in the present study. Notably, however, a dominant protein-protein interface between dimers was not evident. Rather, linkage between the membrane-associated regions could be detected at many positions, arguing for a disordered arrangement. As our single-cysteine scanning approach can only examine the homotypic pairing of cysteine residues, it remains possible that heterotypic pairing of cysteine substitutions may detect a more complex protein interface between dimers, as seen for linkage at the Bak BH3:groove interface (M71C:K113C, *Figure 4B*) (*Dewson et al., 2008*). However, in support of the notion that there is no dominant protein interface between dimers, the interaction between Bak dimers is lost when digitonin is added, in contrast to the BH3:groove interface which persists in the presence of digitonin. Moreover, the lability of dimer-dimer association was highlighted by the capacity of Bax to intermingle with pre-formed Bak dimers. Each of Bcl$_{xL}$, Mcl-1 and Bax localised to mitochondria with similar efficiency, however a critical feature of Bax is that, like Bak, following activation by tBid it collapses onto the membrane surface to form homodimers (*Westphal et al., 2014*). The large membrane surface occupied by Bax dimers,

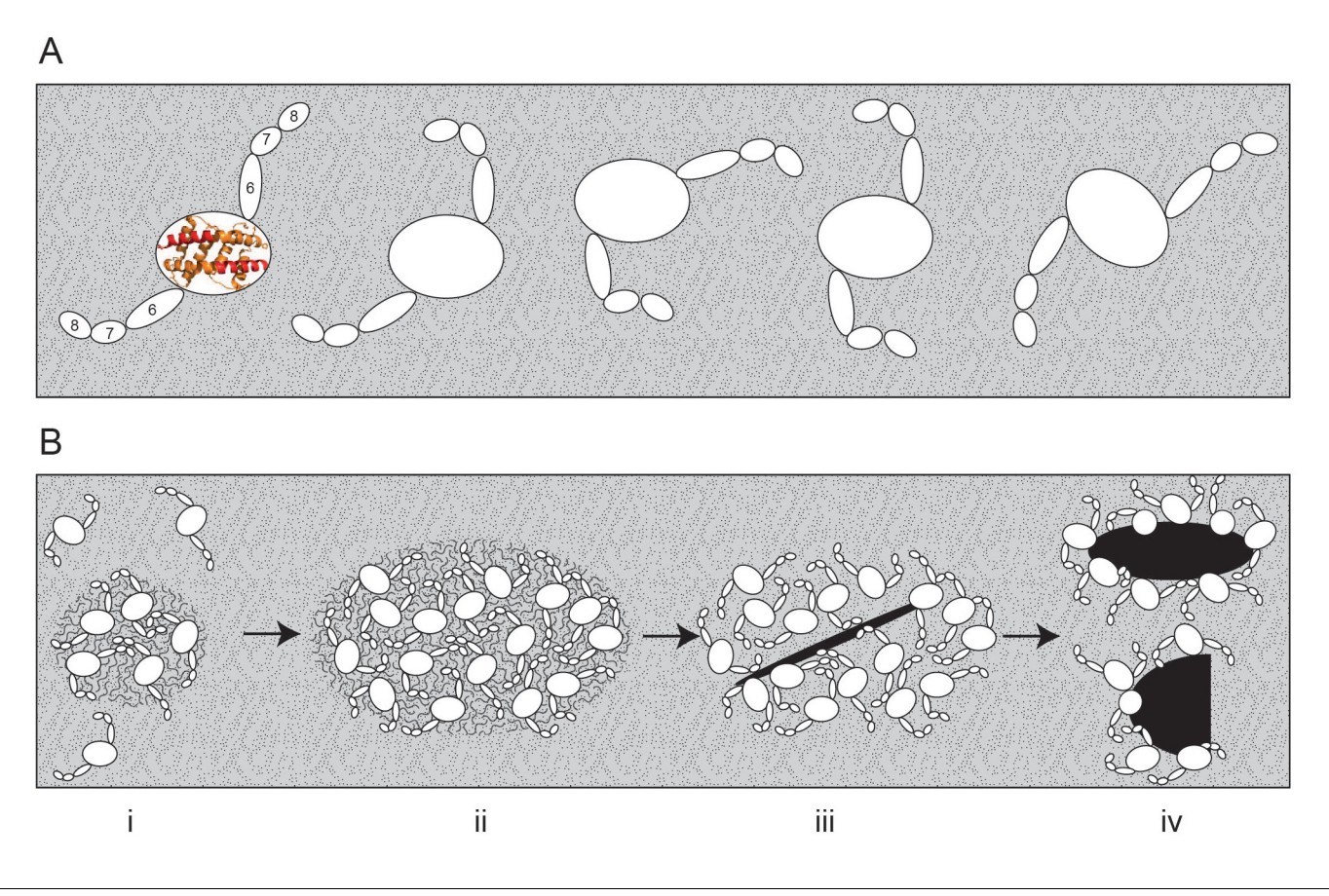

**Figure 7.** Bak dimers adopt various conformations on the membrane surface and aggregate in compact, disordered clusters to disrupt the mitochondrial outer membrane. (**A**) The top view of Bak dimers lying in-plane on the mitochondrial surface. The α2-α5 core is bounded by the large oval. Extending from the core are the membrane anchored C-terminal helices α6, α7 and α8. For simplicity, the α9 transmembrane domains that project into the membrane plane, and the flexible, solvent exposed N-termini of each dimer are not shown. (**B**) We hypothesise that growing clusters of Bak dimers induce membrane tension to rupture mitochondria. (i) Upon activation, Bak dimers penetrate the outer leaflet of the membrane and accumulate in a compact irregular cluster. (ii) More dimers converge on the cluster thus enlarging the patch of membrane disturbance. (iii) Once the patch attains a critical area a non-lamellar lipidic arrangement is generated, relieving membrane tension. (iv) Lipids and Bak dimers rearrange to bury exposed hydrophobic surfaces, yielding a variety of proteolipid (toroidal) 'pores'. Our assays survey a mixture of Bak linkage products derived from stages (ii), (iii) and (iv).

together with disturbance of the outer leaflet, may explain why Bax, but not the prosurvival proteins, could disrupt Bak dimer-dimer association in this assay.

A variety of biophysical and imaging techniques have been deployed to better understand how Bak (or Bax) dimers associate, yet it remains unclear. Well-characterised pore-forming proteins typically reveal an extensive interface between subunits that facilitates pore formation (*Song et al., 1996*; *Mueller et al., 2009*). Versions of this regular architecture have recently been considered for Bak and Bax as a closed circuit of dimers lining a circular pore (*Aluvila et al., 2014*; *Bleicken et al., 2014*; *Brouwer et al., 2014*). In contrast to these hypothetical linear dimer arrangements, our biochemical data revealed that there is no single dominant interface between the α2-α5 core dimers and support the random collision of Bak dimers in two dimensions at the mitochondrial outer membrane during apoptosis. Here a novel approach of mathematical modelling was used to simulate linkage between Bak dimers and showed the feasibility of random dimer association. For example, as the V61C residue linked only between Bak dimers, and did so very efficiently, this provided a powerful tool for testing different arrangements of Bak dimers on the membrane. A minimalist set of assumptions were adopted for stochastic simulations of V61C:V61C' linkage between Bak dimers:

linkage is complete and irreversible but occurs only between dimers; only two linkages are possible per dimer; and neighbouring dimers can only link if both have a free cysteine available for disulphide bonding. Indeed, the simulation showed good concordance with different sets of linkage data obtained from fully oligomerised Bak in mitochondria. From this, we conclude that Bak dimers are closely packed and aggregate with a random orientation.

While our data indicate that most Bak (and probably Bax) oligomers are disordered clusters, it remains possible that a sub-population of ordered oligomers are responsible for driving pore formation. However, in our model system, preventing ~50% Bak from locating to mitochondria prevented pore formation (*Ferrer et al., 2012*) argues that, of the whole population of Bak surveyed in our linkage studies, at least 50% is required for a pore to form somewhere in the mitochondrion. Thus, the patterns observed in our linkage assays are at least 50% representative of Bak engaged in driving pore formation.

A disordered cluster of dimers is reminiscent of the 'carpet' model of pore formation by peptides (reviewed by *Gilbert [2016]*). In this model, amphipathic antimicrobial peptides such as melittin lie parallel to the membrane plane, with accumulation causing strain in the outer layer of the membrane until at higher concentrations the lamellar structure of the membrane is destabilised and non-lamellar lipidic pores form (*Lee et al., 2008*, *2013*). Like an amphipathic peptide, the Bak and Bax α2-α5 core dimers and α6-helices penetrate the outer leaflet of the bilayer (*Czabotar et al., 2013*; *Aluvila et al., 2014*; *Brouwer et al., 2014*; *Westphal et al., 2014*). While very high concentrations of antimicrobial peptides can disintegrate membranes in a detergent-like manner (*Lee et al., 2008*), this does not appear to occur with near-physiological levels of Bak and Bax. Antimicrobial peptides also have membrane thinning attributes (*Chen et al., 2003*), as reported for Bax (*Satsoura et al., 2012*). Satsoura and colleagues discuss how membrane deformation by Bax may promote oligomerisation mediated not by protein:protein interactions but by long-range changes in membrane tension (*Satsoura et al., 2012*). For example, the attractive forces between membrane wrapped particles can arise purely as a result of membrane forces (*Reynwar et al., 2007*; *van der Wel et al., 2016*; *Katira et al., 2016*). Thus, as illustrated in *Figure 7B*, we hypothesise that Bak dimers penetrate the outer leaflet to attract further dimers and in doing so increase the membrane disturbance. Enlarging clusters then destabilise the membrane sufficiently to form lipidic pores that release apoptogenic factors. The pores may then be stabilised by parts of the Bak and Bax dimers rearranging to line the pore (*Terrones et al., 2004*; *García-Sáez, 2012*; *Westphal et al., 2014*; *Kuwana et al., 2016*; *Mandal et al., 2016*), consistent with the lipidic pore model first proposed for Bax (*Basañez et al., 1999*). Hence, our model for the disordered clustering of Bak dimers provides a holistic molecular explanation for the detection of Bak and Bax in a variety of formations including complete rings, arcs, lines and clusters (*Große et al., 2016*; *Nasu et al., 2016*; *Salvador-Gallego et al., 2016*).

In conclusion, this study proposes a novel means of oligomerisation by Bak and Bax in apoptotic cells: that dimers aggregate in dense clusters without a dominant interface and this dynamic association requires the involvement of the lipid membrane. We have described molecular tools that can precisely monitor the aggregation of Bak dimers in cells, and could also be used to examine how the pro-survival proteins inhibit Bak oligomerisation, which may impact on the development of cancer therapies that target these proteins. Critically, the combination of linkage data and mathematical simulations described here has offered, for the first time, insight into the assembly of the apoptotic pore with single molecule resolution.

## Materials and methods

### Recombinant proteins

Recombinant proteins were prepared as previously described; caspase 8-cleaved human Bid (tBid) (as per [*Kluck et al., 1999*]), tBid[Bax] and Bcl-x[L] full length (as per [*Hockings et al., 2015*]), Cys null Bax full length (as per [*Czabotar et al., 2013*]), Mcl-1ΔN151ΔC23 (as per (*Chen et al., 2005*). tBid[M97A] was kindly provided by E. Lee (*Lee et al., 2016*).

## Bak variant library cloning and expression

A library of human Bak (or Bax) mutants was generated by site-directed cysteine substitution (as per (*Dewson et al., 2008*). Human Bak (or Bax) with the endogenous cysteine residues mutated to serine (Bak Cys null C14S/C166S, or Bax Cys null C62S/C126S S184L) was used as the template in overlap extension PCR to introduce single cysteine residues throughout the Bak or Bax sequence (primer sequences listed in *Supplementary file 1*), and then cloned into the pMX-IRES-GFP retroviral vector (primer and vector sequences available on request). Mutants were introduced into SV40-immortalised *Bak*$^{-/-}$ *Bax*$^{-/-}$ mouse embryonic fibroblasts (MEFs) by retroviral infection (as per (*Dewson et al., 2008*). *Bak*$^{-/-}$ *Bax*$^{-/-}$ MEFs have been described previously (*Wei et al., 2001*). Cell lines were checked for mycoplasma contamination.

## Preparation of membrane fractions enriched for mitochondria

*Bak*$^{-/-}$ *Bax*$^{-/-}$ MEFs expressing Bak cysteine mutants at levels comparable to endogenous mouse Bak (*Alsop et al., 2015*) were permeabilised and membrane fractions containing the mitochondria were isolated as previously described (*Dewson et al., 2008*). Cells were washed in PBS and then resuspended at a concentration of $1 \times 10^7$ cells/ml in wash buffer supplemented with digitonin (100 mM sucrose, 20 mM HEPES-NaOH pH 7.5, 100 mM KCl, 2.5 mM MgCl$_2$, 4 µg/ml Pepstatin A (Sigma-Aldrich, St. Louis, MO, USA), Complete protease inhibitors (Roche, Castle Hill, NSW, Australia) and 0.025% w/v digitonin (Calbiochem, Merck, Darmstadt, Germany). Cells were incubated for 10 min on ice, then membrane fractions were collected by centrifugation at 16,000 $\times g$ for 5 min and the supernatant discarded. Pellets were re-suspended in wash buffer (no digitonin), and permeabilisation of the cell membrane confirmed by trypan blue uptake.

## Activation of Bak by tBid or etoposide

To assess the apoptotic function of each Bak variant in cells, *Bak*$^{-/-}$*Bax*$^{-/-}$ MEFs expressing Bak cysteine mutants were treated with etoposide (10 µM) for 24 hr, and cell death, as indicated by propidium iodide (5 µg/ml) uptake, determined by flow cytometry (FACSCalibur, BD Biosciences, San Jose, CA, USA). To activate Bak and induce mitochondrial outer membrane permeabilisation in vitro, membrane fractions from *Bak*$^{-/-}$*Bax*$^{-/-}$ MEFs expressing Bak cysteine variants were treated with 100 nM recombinant tBid for 30 min at 30°C (as described in *Dewson et al., 2008*). Selected samples were treated with the functionally equivalent protein tBid$^{Bax}$ in which the BH3 domain of Bid was substituted with the Bax BH3 domain (*Hockings et al., 2015*). An alternate stimulus was also performed in which membrane fractions were heated at 43°C for 30 min. Release of cytochrome *c* was assessed by centrifugation 16,000 $\times g$ for 5 min to separate pellet and supernatant fractions and samples analysed by reducing SDS-PAGE, followed by immunoblotting with anti-cytochrome *c*.

## Cysteine accessibility to IASD labelling, one-dimensional isoelectric focussing and IASD quantification

The cysteine residue of each Bak variant was assessed for solvent exposure by incubating membrane fractions with the cysteine-labelling reagent IASD (Molecular Probes Life Technologies, Carlsbad, CA, USA) for 30 min at 30°C (as described in [*Westphal et al., 2014*]). Membrane fractions were supplemented with 100 µM TCEP to prevent oxidisation of cysteines that would inhibit IASD labelling. Samples were unlabelled, or incubated with IASD before, during or after tBid incubation or following denaturation with 1% w/v ASB-16 (Merck). IASD labelling was quenched by the addition of 200 mM DTT and samples solubilised in 1% ASB-16 for 10 min at 22°C. Soluble supernatant fractions were isolated by centrifugation at 16,000 $\times g$ for 5 min, and added to an equal volume of IEF sample buffer (7 M urea, 2 M thiourea, 2% w/v CHAPS, Complete protease inhibitors, 4 µg/ml pepstatin A, 1% w/v ASB-16 and 0.04% w/v bromophenol blue). Samples were loaded onto 12-well Novex pH 3–7 IEF gels (Life Technologies) and focused with a Consort EV265 power supply with increasing voltage (100 V for 1 hr, 200 V for 1 hr and 500 V for 30 min). IEF gels were pre-soaked in SDS buffer (75 mM Tris/HCl, pH 6.8, 0.6% w/v SDS, 15% v/v glycerol), and then transferred to PVDF membrane for western blot analysis.

IASD labelling was quantified (ImageLab 4.1, Bio-RAD) from western blots by measuring signal intensity for the unlabelled and labelled Bak band species in each lane to calculate the percentage of the total signal from each lane attributed to the faster migrating IASD-labelled species. Control

samples showed no IASD labelling of Bak Cys null (*Westphal et al., 2014*; *Iyer et al., 2015*). A global background subtraction was applied to regions of interest from each western blot. Data were presented as the mean ± SD (n ≥ 3), or range (n = 2) and the number of replicates indicated on the x-axis. A two-tailed unpaired t-test was employed to determine significant differences (p<0.05) in the percentage of IASD labelled Bak before versus after tBid treatment.

## Oxidant-induced disulphide linkage (CuPhe)

The cysteine residue of each Bak variant was tested for disulphide linkage with proximal cysteine residues on Bak (or other proteins) by incubation with the oxidant copper phenanthroline (CuPhe) (as per [*Dewson et al., 2008*]). CuPhe was prepared as a stock solution of 30 mM $CuSO_4$ and 100 mM 1,10-phenanthroline in 4:1 water/ethanol. Following incubation of membrane fractions with tBid at 30°C, samples were pre-chilled on ice for 5 min, and then incubated with a 100-fold dilution of the CuPhe solution on ice for 15 min. Note that (*Iyer et al., 2015*) performed CuPhe linkage without the pre-chill incubation. Disulphide bond formation (linkage) was quenched by the addition of 20 mM N-ethyl maleimide (NEM, to label any remaining free cysteines, 10 min on ice) and 5 mM EDTA (to chelate copper, 5 min on ice) and samples analysed by non-reducing SDS PAGE or blue native PAGE (BNP).

## SDS PAGE and blue native PAGE

CuPhe-linked samples were analysed by SDS PAGE (12% TGX gels (Life Technologies)) in the absence of reducing agents to preserve disulphide bonds. The efficiency of linkage between Bak molecules was measured by a shift in Bak migration from 1x to 2x complexes. To discriminate between linkages occurring within or between Bak dimers, CuPhe-linked samples were also analysed in tandem by blue native PAGE (BNP) to preserve the native dimer interface (i.e. the BH3:groove interface) in addition to any disulphide bonds between Bak molecules. Migration of higher order linked complexes (i.e. greater than dimer) on BNP indicated the presence of disulphide bonds between Bak dimers. Following quenching of CuPhe with NEM and EDTA, membrane fractions were isolated by centrifugation at 16,000 $\times g$ for 5 min. Supernatants were discarded and membrane pellets were solubilised in 20 mM Bis-Tris pH 7.4, 50 mM NaCl, 10% v/v glycerol, 1% w/v digitonin, and incubated on ice for 1 hr and insoluble material was removed by centrifugation at 16,000 $\times g$ for 5 min. The resulting supernatants were prepared for BNP by the addition of Native Sample buffer (Life Technologies) and Coomassie Additive (Life Technologies), and then loaded onto Novex 4–16% Native PAGE 1.0 mm 10 well gels as per the manufacturer's instructions (Life Technologies). Western blot transfer to PVDF was performed at 30 V for 150 min in transfer buffer (25 mM Tris, 192 mM Glycine, 20% v/v Methanol) supplemented with 0.037% w/v SDS. PVDF membranes were de-stained with 10% v/v acetic acid 30% v/v ethanol, then further de-stained in methanol and rinsed thoroughly with $dH_2O$ before immunoblotting.

## Immunoblotting

SDS PAGE and IEF membranes were immunoblotted for Bak using the rabbit polyclonal anti-Bak aa23-38 (B5897, Sigma-Aldrich, Castle Hill, NSW, Australia, RRID:AB_258581). BNP membranes were immunoblotted for Bak using an in-house anti-Bak monoclonal rat IgG (clone 7D10, WEHI, [*Dewson et al., 2009*; *Alsop et al., 2015*]), except in the case of Bak mutants ranging from G51C to P58C that were detected on BNP with the anti-Bak aa23-38. In-house monoclonal rat antibodies were used to detect recombinant Bax (clone 49-F9, WEHI), Bcl-$x_L$ (clone 9C9, WEHI) and Mcl-1 (clone 19C 4–15, WEHI). Cytochrome *c* release was assessed by SDS-PAGE and immunoblotting with anti-cytochrome *c* (clone 7 H8.2C12, 556433, BD Biosciences Pharmingen, San Diego, CA, USA RRID:AB_396417). Horseradish peroxidase conjugated IgG secondary antibodies were used; anti-rabbit (4010–05, AdB Serotec, RRID:AB_609701), anti-rat (3010–05, AdB Serotec, RRID:AB_619911) and anti-mouse (1010–05, AdB Serotec, RRID:AB_609673). Immobilised horseradish peroxidase was detected with Luminata Forte western HRP substrate (WBLUF0500, Millipore, Billerica, MA, USA), images captured with the ChemiDoc XRS+ System (Bio-RAD, Hercules, CA, USA) and signal intensity measured with Image Lab 4.1 software (Bio-RAD).

## Blue native PAGE densitometry

Images were transformed in Image Lab 4.1 (Bio-RAD) to correct any rotation of the image and exported as.tif for densitometry in FIJI (ImageJ 1.47n) (*Schindelin et al., 2012*; *Schneider et al., 2012*). The lookup table (LUT) was inverted and images rotated 90 degrees. Using the rectangular selection tool, a box was drawn around the lane of interest to encompass the full range of Bak signal (with fixed dimensions of 370 $\times$ 50 pixels). This box was then measured with the Dynamic ROI profiler plugin. Plot values were exported for analysis in Prism (ver 6.0f, Graphpad Software Co., La Jolla, CA, USA), and normalised to a fraction of the total intensity for each lane of interest. A clearer representation of multiple replicate plots of lane intensity was achieved by a linear rescaling of each replicate's horizontal axis so that the maximum corresponding to Bak 4x species, and the characteristic minimum (which is a highly reproducible artefact of our Bak BNP western blots) at ~720 kDa, coincided across replicates.

## Simulation of linkage between dimers (V61C:V61C')

The formation of a disulphide linked oligomer is modelled by repeatedly (a) selecting an available pair of cysteines on neighbouring dimers at random, (b) declaring them linked and (c) removing those cysteines from the list of those subsequently available. We continue until no neighbouring dimers have cysteines available for linkage, to reflect complete linkage of V61C:V61C'. We consider a collection of Bak dimers on a two-dimensional surface (see *Figure 6*). Dimers are arranged on a hexagonal (or square) grid, and linkage is permitted in all directions. Each dimer has two cysteines available for disulphide linkage to a neighbouring dimer. We simulate such that linkage is allowed in all directions with equal probability. Reduced linkage efficiency was modelled by introducing an edge blocking effect. Every edge between neighbouring hexagonal cells was flagged as eligible for a disulphide bond, independently at random with probability $p$ constant across all edges in a given simulation, for p=0, 0.25, 0.5 and 0.75. We modelled the experimental scenario where Bak dimers were formed from a mixture of equal amounts of V61C and cys-null monomers by assigning each dimer either zero, one or two available cysteines with probability 0.25, 0.5 and 0.25 respectively (see *Figure 6—figure supplement 3C*).

Predicted densitometry for each Bak dimer multiplicity are presented by taking the distribution of oligomer sizes on completion of the simulation, multiplying by the number of Bak dimers in each oligomer to obtain the theoretical density that would give rise to in the western blot, using an exponential horizontal axis to mimic the nonlinear spacing of bands, and applying a gaussian kernel estimator to approximate the smearing of bands and their running into one another at higher molecular weights (see *Figure 6—figure supplement 1*).

## Acknowledgements

The authors declare no conflict of interest. We thank Peter Colman, Peter Czabotar, John Markham, Matthew Ritchie, Aline Oelgeklaus and Sheilah Gaines for critical comments on the manuscript. We thank Colin Hockings for helpful discussions regarding simulations. We gratefully acknowledge the IASD labelling assay development and analysis methods of Dana Westphal. Our work is supported by NHMRC grants (637337 and 1016701), and the Victorian State Government Operational Infrastructure Support and the Australian Government NHMRC IRIISS.

## Additional information

### Funding

| Funder | Grant reference number | Author |
| --- | --- | --- |
| National Health and Medical Research Council | Project grant 637337 | Ruth M Kluck |
| State Government of Victoria | Operational Infrastructure Support | Ruth M Kluck |
| National Health and Medical Research Council | Program grant 1016701 | Ruth M Kluck |

The funders had no role in study design, data collection and interpretation, or the decision to submit the work for publication.

## Author contributions

RTU, Conceptualization, Formal analysis, Supervision, Investigation, Visualization, Methodology, Writing—original draft, Writing—review and editing; MO'H, Data curation, Software, Formal analysis, Visualization, Methodology, Writing—review and editing; SI, Investigation, Visualization, Methodology, Writing—review and editing; RB, MXS, JMB, AEA, Investigation, Writing—review and editing; GD, Conceptualization, Investigation, Methodology, Writing—review and editing; RMK, Conceptualization, Supervision, Visualization, Methodology, Writing—original draft, Writing—review and editing

## Author ORCIDs

Rachel T Uren, http://orcid.org/0000-0001-7456-478X

Martin O'Hely, http://orcid.org/0000-0002-0212-1207

Sweta Iyer, http://orcid.org/0000-0002-7787-5118

Melissa X Shi, http://orcid.org/0000-0003-1454-9566

Amber E Alsop, http://orcid.org/0000-0001-7894-7294

Grant Dewson, http://orcid.org/0000-0003-4251-8898

Ruth M Kluck, http://orcid.org/0000-0002-7101-1925

# Additional files

## Supplementary files

• Supplementary file 1. Primer sequences (5'−3') used for PCR mutagenesis.

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
