## [Decision Letter]

Thank you for submitting your article "Disordered clusters of Bak dimers rupture mitochondria during apoptosis" for consideration by *eLife*. Your article has been favorably evaluated by John Kuriyan as the Senior Editor and three reviewers, one of whom is a member of our Board of Reviewing Editors. The reviewers have opted to remain anonymous.

The reviewers have discussed the reviews with one another and the Reviewing Editor has drafted this decision to help you prepare a revised submission.

Summary:

Uren et al. investigate structural rearrangements associated with the oligomerization of BAK during MOMP, which is an important and debated research question in the apoptosis field. Following previous work, the authors used an expanded set of monocysteine BAK mutants in IASD labeling and cysteine crosslinking experiments to extract new information about the topology and multimerization surfaces of cBID-activated BAK. In addition, the authors used a novel approach of mathematical modeling to simulate linkage between BAK dimers and showed feasibility of random dimer association into high-order oligomers. Based on these collective approaches, Uren et al. elaborate a model where BAK dimers aggregate into disordered clusters lacking a dominant protein-protein interface to elicit MOMP.

Essential revisions requiring new experimental work:

Considering the information already available for BAX/BAK membrane topologies, IASD labelling data appears rather incomplete (Figure 1). First of all, following previous work by this and other groups it should be straightforward to discriminate between membrane-inserted and protein buried residues, at least for a few key residues (see below). Second, the author´s conclusion that the entire BAK helix 1 becomes solvent-exposed upon BAK activation primarily relies on the finding that several hydrophilic sites within BAK helix 1 (R36, Y41, Q44 and Q47) show increased IASD labelling after cBID treatment. To firmly establish this point, the IASD labelling test should be extended to additional hydrophobic sites in the BAK helix 1 (such as V27, F35, V39, and F40), because these hydrophobic sites display the highest potential for becoming buried in the hydrophobic interior of the membrane or a proteinaceous structure. Similarly, additional hydrophobic sites in BAK helix 2 (I81, I85) and BAK helix 4 (F111, T112) should also be tested for IASD labelling to unambiguously establish that cBID-activated BAK adopts a BH3-into-groove dimeric structure at the MOM, as the authors claim. This is also critical, because Zhang et al. recently reported that active BAX adopt an helix 2-helix 3-helix 4 dimeric structure rather than a BH3-into-groove structure at the MOM by analysing analogous sites in BAX helix 2 (I66, L70) and BAX helix 4 (F92, F93). In addition to BAK helix 1, extending the IASD-labelling studies to more hydrophobic sites in the BAK helix 7-8 region also appears highly desirable.

2) To establish the generality of the findings obtained with the BAK V61C mutant, it is important to show that the same crosslinking pattern elicited by cBID is reproduced by other apoptotic stimuli such as heat (43°C), etoposide (+QVD-oph), and/or BIMs overexpression (+QVD-oph). Also, have the authors excluded the possibility that VDAC2 or BCLXL molecules are present in the high-order BAK V61C multimeric bands detected by BN-PAGE?

Other important points to consider (not necessarily requiring experimental work:

1A) One confusing aspect is why these assemblies of Bax and Bak don't diffuse in the plane of the membrane throughout the mitochondrial surface. There must be some affinity among the dimers to keep them near enough to interact, appear focal in imaging studies and to destabilize the membrane (as shown by the authors in Figure 7Bi and ii). And "some affinity" likely means a protein protein interface. It is hard to understand how the Bak to mitochondria ratio used in this study relates to in vivo situations. If it is too high, non-specific non-physiologic interactions may mask the specific interactions that occur at endogenous levels of protein.

1B) Would consideration of possible non-symmetric dimer-dimer contact sites (i.e. linkage of different cys), as the authors brought up, challenge the authors strong conclusion of "no single dominant interface between the α2-α5 core dimers"? Please rebut these points or tone down the related statements in the manuscript.

2) In the last paragraph of the subsection “Mobility of the N- and C-extremities permits linkage between dimers”, where the data is discussed regarding the BNP ladder after crosslinking Bak with C-terminal helix cysteines – it would be valuable to add in another scheme into Figure 3 that depicts what the authors mean by α6-α8 shielding of regions of helix 9.

3) In Figure 4: is the tBID removed from the Bak and membranes prior to adding the Bax and Bcl-x_L_? If not, tBID activating Bax or potentially inhibiting Bcl-x_L_ confounds the interpretations. I realize the authors use a mutant tBID to minimize this worry but the fact that Bcl-x_L_ and Mcl1 translocate to mitochondria, as they do in cells upon BH3 only protein induction, suggests it may be a problem even for these anti-apoptotic proteins. If removal of tBid is impractical, why not activate Bak with the 7D10 antibody used previously by the Kluck group? It would be interesting to compare Bax and Bcl-x_L_ activity on Bak oligomers activated by 7D10 with and without added tBid.

4) As it stands now, the manuscript presents very limited experimental evidence supporting the notion that disordered BAK clusters directly cause MOMP, rather than merely an epiphenomenon that accompanies MOMP. This is a critical issue that needs to be further substantiated by experimental evidence or at least their claim that BAK clusters mediate MOMP, rather than merely accompanying MOMP, toned down.

5) As in any similar type of approaches where one uses mutations to probe conformation, the conclusion from the observations hinges on two main assumptions: 1) The conformation of the protein itself has not been altered by the mutation to incorporate cysteine, or modification of the introduced cysteine by a chemical compound (such as IASD) has not triggered additional conformational change; 2) The physical properties of the modified protein has not changed so that the authors are not observing non-physiological events. I do understand that in some of these studies experimental controls to the above assumptions may not be easy or possible, but it would be useful if the authors can at least attempt at considering them. Testing apoptotic function of the mutant Bak in cells is a good start. Especially those mutations that involved hydrophobic and buried residues.

6) How did the authors determine that the linkages for some mutants were to other proteins rather than Bak? The authors suggested that this is due to close proximity on the mitochondria membrane and not due to potential physical interaction. It could be argued that the potential interaction with other proteins can't be completely ruled out in the linkage bands observed.

---

## [Author Response]

*Essential revisions requiring new experimental work:*

*Considering the information already available for BAX/BAK membrane topologies, IASD labelling data appears rather incomplete (Figure 1). First of all, following previous work by this and other groups it should be straightforward to discriminate between membrane-inserted and protein buried residues, at least for a few key residues (see below).*

Discriminating between membrane-inserted and protein buried residues is unfortunately not straightforward. The IASD labelling data (Figure 1) uses the IASD moiety alone to discriminate between exposed and buried residues. Accordingly, our previous studies in Bak and Bax (Westphal et al., PNAS 2014) confirmed that IASD labelling alone labels solvent exposed residues (Zhu Q & Casey JR, Methods 2007).

To discriminate whether buried residues are buried in protein or membrane, as the reviewer notes, others have found that treatment with urea specifically disrupts protein:protein interactions and CHAPS specifically uncovers membrane buried residues for Bax. Unfortunately, in our hands the use of urea and CHAPS has been unreliable for Bak (not shown).

*Second, the author´s conclusion that the entire BAK helix 1 becomes solvent-exposed upon BAK activation primarily relies on the finding that several hydrophilic sites within BAK helix 1 (R36, Y41, Q44 and Q47) show increased IASD labelling after cBID treatment.*

As the reviewer notes, the conclusion that the entire BAK helix 1 becomes solvent-exposed upon Bak activation is supported by increased solvent exposure of R36C, Y41C, Q44C and Q47C (and new data also reveals a trend for increased exposure of V39C). However, we also show that 30 other residues tested (out of a total of 70) in the Bak N-extremity were solvent-exposed both before and after activation. That is, no residues tested in the N-extremity become buried (35 residues summarised in Figure 1). As discussed in the text, extensive exposure of the N-terminus is also consistent with a recent study from our group (Alsop et al., Nat Commun 2015) showing exposure of several N-terminal antibody epitopes following activation of Bak (each of which recognised linear epitopes). In addition, several studies have shown that this region becomes increasingly sensitive to protease digestion upon activation. Thus, several approaches argue that the Bak N-terminus becomes fully exposed upon activation.

*To firmly establish this point, the IASD labelling test should be extended to additional hydrophobic sites in the BAK helix 1 (such as V27, F35, V39, and F40), because these hydrophobic sites display the highest potential for becoming buried in the hydrophobic interior of the membrane or a proteinaceous structure.*

As requested, we have conducted IASD labelling for additional hydrophobic sites in Bak helix 1 (F35C, V39C, F40C). We find that V39C shows a trend for increased solvent exposure upon activation, whereas F35C and F40C are solvent exposed throughout. These data agree with our conclusion that the Bak N-terminus is fully exposed upon activation. These new data have been incorporated into Figure 1; Figure 1—figure supplement 3 and [Supplementary-material SD1-data] . The second paragraph of the subsection “The Bak N-segment, α1 and α1- α2 loop are solvent-exposed in Bak oligomers” now incorporates the new data, and clarifies evidence for full exposure of the N-terminus.

*Similarly, additional hydrophobic sites in BAK helix 2 (I81, I85) and BAK helix 4 (F111, T112) should also be tested for IASD labelling to unambiguously establish that cBID-activated BAK adopts a BH3-into-groove dimeric structure at the MOM, as the authors claim. This is also critical, because Zhang et al. recently reported that active BAX adopt an helix 2-helix3-helix4 dimeric structure rather than a BH3-into-groove structure at the MOM by analysing analogous sites in BAX helix 2 (I66, L70) and BAX helix 4 (F92, F93).*

We note that the helix 2-3-4 dimeric Bax structure reported by Zhang et al. appeared to be a subpopulation of Bax, with BH3-into-groove structures also present and the physiological importance of the alternate 2-3-4 dimeric Bax is not yet established.

We also note that Bak variants I81T, I85T and F111S cannot mediate apoptosis in cells and fail to homodimerise (Dewson et al., Mol Cell 2008). We have since generated the Bak I85C variant and have found that this variant is also non-functional. We did not pursue the generation of cysteine substitutions for I81 or F111 and we have not tested mutation of position T112.

However, we can now show that our findings with linkage at Bak V61C are generalisable to Bax. We have created the equivalent of the Bak V61C mutation in membrane localised Bax (Bax A46C/S184L) (Figure 3—figure supplement 1). This Bax variant can be linked between (but not within) dimers, indicating that Bax dimers have flexible N-terminal extremities and a constrained core with a topology similar to the Bak dimer. Additional text has been added in the third paragraph of the subsection “Mobility of the N- and C-extremities permits linkage between dimers”.

*In addition to BAK helix 1, extending the IASD-labelling studies to more hydrophobic sites in the BAK helix 7-8 region also appears highly desirable.*

We have already presented IASD labelling data for cysteine substitutions in helix 7 (Q173C) and helix 8 (V178C, A179C, L181C, N182C) in Figure 1 and Figure 1—figure supplement 3 (where Bak was activated by tBid^BaxBH3^). We now include data for Q173C in Figure 1 and Figure 1—figure supplement 3.

We also generated an additional variant at a hydrophobic site in Bak helix 7 (W170C). Unfortunately, expression of Bak W170C in Bak^-/-^ Bax^-/-^ MEFs was very low and accordingly mitochondrial fractions from these cells did not release cytochrome *c* in response to tBid treatment. As there was no evidence that this is a functional Bak variant, IASD labelling analysis was not performed.

*2) To establish the generality of the findings obtained with the BAK V61C mutant, it is important to show that the same crosslinking pattern elicited by cBID is reproduced by other apoptotic stimuli such as heat (43°C), etoposide (+QVD-oph), and/or BIMs overexpression (+QVD-oph).*

As requested, we present a comparison of tBid treatment with heat treatment (43°C) in our mitochondrial cysteine linkage assay with the Bak V61C mutant (Figure 3—figure supplement 1, compare lanes 4 (tBid treatment) and lane 5 (43°C treatment). The same crosslinking pattern is induced by tBid or heat treatment, arguing that the linkage pattern between dimers is generalisable not only to both Bak and Bax, but to the two activators tBid and heat. Additional text has been added in the third paragraph of the subsection “Mobility of the N- and C-extremities permits linkage between dimers”.

*Also, have the authors excluded the possibility that VDAC2 or BCLXL molecules are present in the high-order BAK V61C multimeric bands detected by BN-PAGE?*

We cannot exclude the possibility that VDAC2, Bcl-x_L_, and other molecules may be in some of the complexes. However, we note that the molecular weights of VDAC2 (~32 kDa) and Bcl-x_L_ (~26 kDa) differ from that of Bak (~23 kDa), so low-order complexes would be expected to run differently to that of Bak multimers. We note also that in Figure 6—figure supplement 3, where V61C is co-expressed with Bak Cys null, the greatly reduced linkage argues that Bak V61C links mostly to itself. This conclusion has now been noted in the figure legend.

*Other important points to consider (not necessarily requiring experimental work):*

*1A) One confusing aspect is why these assemblies of Bax and Bak don't diffuse in the plane of the membrane throughout the mitochondrial surface. There must be some affinity among the dimers to keep them near enough to interact, appear focal in imaging studies and to destabilize the membrane (as shown by the authors in Figure 7Bi and ii). And "some affinity" likely means a protein protein interface.*

We and others have searched for a protein protein interface between dimers to explain why dimers do not diffuse in the plane of the membrane. However, the present study has failed to identify such an interface, and thus we hypothesise that membrane forces are driving Bak dimer aggregation. This hypothesis is consistent with a reversible attraction occurring between membrane wrapped particles purely as a result of membrane forces (Reynwar et al., Nature 2007, van der Wel et al., Sci Reports 2016, Katira et al., eLife 2016).

We have discussed this concept in more detail in the sixth paragraph of the Discussion. Briefly, our data suggest that when the Bak dimer forms, it adopts an in-plane topology at the membrane and the C-terminal α6 helix becomes partially embedded in-plane with the outer leaflet of the membrane bilayer. Furthermore, the Bak dimer structure (4U2V) has several hydrophobic moieties on the underside of the dimer (helices 4 and 5) thus it is plausible that this dimer would “sink” into the hydrophobic environment of the outer membrane. We hypothesise that this embedded dimer displaces molecules in the outer leaflet, and this deformation induces line tension in the membrane. The membrane is a fluid environment and random movement of Bak dimers on this surface will occur. However, the membrane will drive aggregation of these patches of disturbed membrane surrounding embedded dimers as it seeks to minimise the elevated line tension that occurs with the asymmetric burial of Bak dimers in the outer leaflet of the membrane.

We hypothesise that when this cluster of dimers reaches a critical size and perturbs a sufficient surface area on the membrane, the increased line tension can no longer be accommodated, thus the membrane is ruptured. We predict that the downstream reorganisation of protein and lipids to bury newly exposed hydrophobic surfaces results in the formation of lipidic pores of varying geometry (as observed in Grosse 2016, Nasu 2016, Salvador-Gallego 2016).

*It is hard to understand how the Bak to mitochondria ratio used in this study relates to in vivo situations. If it is too high, non-specific non-physiologic interactions may mask the specific interactions that occur at endogenous levels of protein.*

We agree that greater than physiological levels of Bak may mask the presence of infrequent but perhaps more meaningful physiological interactions. However, in the present studies, the level of re-expressed human Bak is comparable to endogenous mouse Bak protein levels. This was shown in a recent study (Alsop et al., Nat Commun 2015), aided by the use of antibodies that recognise a linear epitope which is identical in mouse and human Bak. This is now clarified in the Methods (subsection “Preparation of membrane fractions enriched for mitochondria”).

*1B) Would consideration of possible non-symmetric dimer-dimer contact sites (i.e. linkage of different cys), as the authors brought up, challenge the authors strong conclusion of "no single dominant interface between the α2-α5 core dimers"? Please rebut these points or tone down the related statements in the manuscript.*

A comprehensive single cysteine mutant screen has not revealed any dominant interface between dimers, and we acknowledge that heterotypic cysteine substitutions in such a screen may have highlighted non-symmetric dimer-dimer contact sites. And those within the highly structured α2-α5 core region would be of most interest in terms of driving oligomers. This is stated in the text (Discussion, third paragraph).

However, our conclusion of no significant interface between the core dimers does not rely entirely on the lack of linkage in the single mutant screen (as discussed in the aforementioned paragraph). That is, detergent can more readily disturb formation of an interface between dimers than the primary BH3:groove interface within dimers (Figure 4), indicating that interactions between dimers are less robust than those within dimers. We have also found that pre-formed aggregates of Bak dimers can be dispersed with the addition of recombinant Bax (Figure 4), thus an energetically favoured protein-protein contact between Bak dimers appears unlikely. In summary, we have used several different approaches to explore the strength of interactions between Bak dimers and have not yet found any evidence of a robust protein interface between Bak dimers.

*2) In the last paragraph of the subsection “Mobility of the N- and C-extremities permits linkage between dimers”, where the data is discussed regarding the BNP ladder after crosslinking Bak with C-terminal helix cysteines – it would be valuable to add in another scheme into Figure 3 that depicts what the authors mean by α6-α8 shielding of regions of helix 9.*

We apologise for the lack of clarity in that paragraph. It has now been re-written (subsection “Mobility of the N- and C-extremities permits linkage between dimers”, last paragraph). The topology is also discussed in further detail in the discussion, with reference to the schematic presented in Figure 7 that shows how the randomly oriented C-terminal helices α6, α7 and α8 on the membrane surface might limit contact between the α2-α5 core region of each dimer and also limit collisions between the transmembrane α9 helices.

*3) In Figure 4: is the tBID removed from the Bak and membranes prior to adding the Bax and* Bcl-x_L_*? If not, tBID activating Bax or potentially inhibiting* Bcl-x_L_*confounds the interpretations.*

In Figure 4, the tBid was not removed, but this does not confound the key conclusion drawn from this data: that Bak dimer clusters are dynamic and can be interspersed with another membrane protein (in this case Bax).

For example, as the reviewer suggests, tBid was not removed so we expect that there is sufficient to activate Bax and to bind to some Bcl-x_L_ and Mcl-1. As the reviewer notes, binding to Bcl-x_L_ and Mcl-1 may occur despite this version of tBid (tBid M97A) binding very poorly to Bcl-x_L_ and Mcl-1 (Supplementary Figure 1A in Lee et al., Genes Dev 2016). However, the most important aspect of Bax, Bcl-x_L_ and Mcl-1 behavior is that a significant portion was able to associate with the membrane fraction, and thus test whether Bak clusters are dynamic. Of these, only Bax was able to intermingle with pre-formed Bak dimers. We did not intend to explore the nuanced interactions between pro-survival proteins and Bak dimers in this work, rather we added Bcl-x_L_ and Mcl-1 as a contrast to recombinant Bax. We were confident that Bcl-x_L_, Mcl-1 and Bax would all localise to the mitochondrial outer membrane, but appreciated that they might have different properties once at the membrane, and this is evident in this assay.

*I realize the authors use a mutant tBID to minimize this worry but the fact that BCl^-^xL and Mcl1 translocate to mitochondria, as they do in cells upon BH3 only protein induction, suggests it may be a problem even for these anti-apoptotic proteins.*

As noted above, we did not intend to explore the nuanced interactions between pro-survival proteins and Bak dimers in this work, rather we added Bcl-x_L_ and Mcl-1 as a contrast to recombinant Bax.

We note however, that the localisation of Mcl-1 and Bcl-x_L_ to the membrane fractions could occur for several reasons. First, as tBid has a membrane anchor, any Mcl-1 or Bcl-x_L_ it bound may also be at the mitochondria, but possibly in a slightly different conformation to the unbound forms. Second, Bcl-x_L_ has the transmembrane domain which may insert without any interaction with other Bcl-2 family members. Third, Mcl-1 and Bcl-x_L_ may bind to activated Bak which is inserted in the membrane, although this population is likely to be minimal as Bak is fully dimerised prior to the addition of Mcl-1 or Bcl-x_L_ (as seen on BNP, e.g. Figure 3—figure supplement 1, lane 2).

*If removal of tBid is impractical, why not activate Bak with the 7D10 antibody used previously by the Kluck group? It would be interesting to compare Bax and* Bcl-x_L_*activity on Bak oligomers activated by 7D10 with and without added tBid.*

Yes, the 7D10 antibody would be an appropriate alternate stimulus for Bak activation, however as discussed, an alternative to tBid is not necessary at this time.

*4) As it stands now, the manuscript presents very limited experimental evidence supporting the notion that disordered BAK clusters directly cause MOMP, rather than merely an epiphenomenon that accompanies MOMP. This is a critical issue that needs to be further substantiated by experimental evidence or at least their claim that BAK clusters mediate MOMP, rather than merely accompanying MOMP, toned down.*

The key data supporting the role of disordered Bak clusters in MOMP is that if we reduce the amount of Bak by 50%, we no longer see pores (Figure 4A in Ferrer et al., PLoS One 2012). Thus, of the whole population of Bak that is surveyed in our linkage studies, at least 50% is required for pore formation. So, the patterns observed in our linkage assays are at least 50% representative of Bak engaged in pore formation. This has been clarified in the fifth paragraph of the Discussion.

As indicated in the Discussion, our current assays allow a range of membrane-perturbing agents to be tested for the ability to inhibit pore formation at the step of dimer-dimer association.

*5) As in any similar type of approaches where one uses mutations to probe conformation, the conclusion from the observations hinges on two main assumptions: 1) The conformation of the protein itself has not been altered by the mutation to incorporate cysteine, or modification of the introduced cysteine by a chemical compound (such as IASD) has not triggered additional conformational change; 2) The physical properties of the modified protein has not changed so that the authors are not observing non-physiological events. I do understand that in some of these studies experimental controls to the above assumptions may not be easy or possible, but it would be useful if the authors can at least attempt at considering them. Testing apoptotic function of the mutant Bak in cells is a good start. Especially those mutations that involved hydrophobic and buried residues.*

We agree that certain assumptions are made when using Bak cysteine substitution variants. As the reviewer suggests, testing the apoptotic function of the mutant Bak proteins in cells is a good start to ensuring data based on those mutants are physiologically relevant. Thus, a cell death assay in response to etoposide has been performed for all variants used in this study, and are reported either in Figure 1—figure supplement 1 (newly generated variants only) or in our previous publications.

Many variants have also been screened in mitochondrial assays for the ability to permeabilise mitochondria in response to tBid. We now include this information as Figure 1—figure supplement 2. In each case, cell killing in response to etoposide correlated with cytochrome c release from membrane fractions treated with tBid.

*6) How did the authors determine that the linkages for some mutants were to other proteins rather than Bak? The authors suggested that this is due to close proximity on the mitochondria membrane and not due to potential physical interaction. It could be argued that the potential interaction with other proteins can't be completely ruled out in the linkage bands observed.*

Linkage between Bak molecules was indicated by the presence of complexes corresponding to twice the molecular weight of the Bak monomer on non-reducing SDS-PAGE. In addition, linkage throughout Bak is a reliable measure of Bak:Bak association, as validated by dimer and oligomer migration on BNP. We note also that it is impressive that our simple computational model of Bak dimers linking (V61C:V61C) only to other Bak dimers shows such good concordance with our empirical BNP data and suggests that linkage to proteins other than Bak is a minor species in the Bak clusters. (Also, see response to Essential Point 2).

We do not rule out linkage to other proteins and the possibility of physiologically relevant interactions between Bak and other proteins. For instance, it has been established that non-activated Bak can associate with VDAC2 in the mitochondrial outer membrane (Cheng E et al., Science 2003; Lazarou M et al., JBC 2010).